# Gut microbiome associations with acute malnutrition relapse in South Sudan

K. Yang,[1] S. King,[2] A. Marshak,[3] L. D'Mello-Guyett,[4] L. Grignard,[5] J. Knee,[4] G. Wong,[1] L. Zhao,[1] N. G. Lamaka,[6] D. Save,[6] M. Gose,[6] A. Myers,[4] I. Trehan,[7,8,9] O. Cumming,[4] H. Stobaugh,[2] D. J. Schwartz[1,10,11,12]

**ABSTRACT** Severe acute malnutrition (SAM) is a leading cause of childhood morbidity and mortality that is defined by anthropometric measurements, weight-for-height $z$ score, and mid-upper arm circumference (MUAC) falling significantly below healthy standards. While treatments for SAM and our understanding of this disease have advanced, children experiencing SAM frequently relapse to acute malnutrition (AM) following anthropometric recovery. Little is known about the contribution of the gut microbiome to AM relapse. We hypothesized that features of the gut microbiome, including microbial composition, antimicrobial resistance gene carriage, and predicted microbial functional pathways, of children discharged from treatment for uncomplicated SAM in South Sudan, may be associated with AM relapse at 1-month follow-up. Overall, broad microbiome profiles at discharge were not associated with AM relapse. We evaluated the associations of microbiome features with AM relapse 1-month post-recovery using mixed linear effect models. We identified associations between higher MUAC, which may be a proxy for future health trajectories, and increased *Sutterella wadsworthensis* and trimethoprim-resistant dihydrofolate reductase antimicrobial resistance genes. These findings suggest that the gut microbiome at discharge of children treated for uncomplicated SAM has limited predictive value as a standalone diagnostic tool for identifying relapse risk at 1 month.

**IMPORTANCE** Severe acute malnutrition (SAM) is a devastating illness that impacts the morbidity and mortality of millions of children worldwide. Community-based management of acute malnutrition (CMAM) is the standard of care in South Sudan and many other low-resource settings for children presenting with SAM. Despite this intervention, children treated for SAM under CMAM frequently relapse to acute malnutrition (AM) following treatment. With advancements in our understanding of malnutrition, there has been a strong and growing interest in developing microbiome-based strategies to treat, prevent, and predict relapse to AM following treatment for SAM. Our work characterizes gut microbiome features of children from a geographic area that is traditionally underrepresented in gut microbiome research and shows that in isolation, a child's gut microbiome at discharge likely holds low predictive value for relapse to AM post-CMAM treatment; however, we identified key microbes and microbial features meriting further research.

**KEYWORDS** CMAM, acute malnutrition in South Sudan, malnutrition relapse, metagenomic sequencing, microbiome, acute malnutrition

Acute malnutrition (AM) is a leading cause of morbidity and mortality in children and affects the broader health and quality of life of families and communities in many low- and middle-income countries (LMICs) (1, 2). In the low-resource setting, accessible and non-invasive anthropometric measurements such as weight-for-height

Address correspondence to D. J. Schwartz, schwartzd@wustl.edu.

The authors declare no conflict of interest.

See the funding table on p. 18.

z score (WHZ) and mid-upper arm circumference (MUAC) are important indices for evaluating the severity of AM (3). WHZ defines the deviation of a child's weight-to-height ratio against un-malnourished children, while MUAC measurements indirectly assess the peripheral distribution of muscle and fat (3). The World Health Organization (WHO) defines AM in children 6–59 months as a WHZ of <−2, an MUAC of <125 mm, or the presence of nutritional bilateral pitting edema, defined as the accumulation of excess fluid in the lower extremity resulting in a lasting skin indentation with applied pressure (3). AM can be further stratified into moderate acute malnutrition (MAM) or severe acute malnutrition (SAM). SAM is associated with greater childhood morbidity and mortality and is defined by more severe anthropometric standards of WHZ <−3, MUAC of <115 mm, or the presence of edema (3).

Community-based management of acute malnutrition (CMAM) is the WHO-endorsed standard of care for treating uncomplicated SAM (3, 4). The treatment regimen includes specially-formulated foods and antimicrobial therapy, namely amoxicillin in South Sudan, given in outpatient settings (3). Despite the high effectiveness of the treatment at curing children to a point of anthropometric recovery, post-treatment relapse to AM is common, ranging up to 37% within 6 months of initial recovery in many LMICs, to as high as 63% in South Sudan (5–12). Several factors have been associated with relapse, including having relatively lower anthropometry at discharge, lower wealth status, and greater severity of household food insecurity (5, 13, 14). Higher incidences of illness and diarrhea occur more frequently in the post-discharge period of children who relapse to AM compared to children who sustain recovery (5). Despite these associations, discordant rates of relapse within children experiencing these risk factors remain high and vary by context (5, 7, 8). This discrepancy highlights the need to more accurately predict which children may be susceptible to relapse and effectively intervene in low-resource settings.

The disruption of the gut microbiome is associated with childhood AM. Children accrue increasing amounts of diverse bacteria, viruses, and fungi in their gut microbiome as they age through their diet and environmental exposures, including in those who have been treated for SAM (15–19). Bacteria in the gut microbiome play a crucial role in food metabolism and nutrient extraction, as well as immune education and protection against enteric pathogens (16, 20). The microbiome also serves as a reservoir for antimicrobial resistance genes (ARGs), which can affect the efficacy of antibiotics such as amoxicillin administered during CMAM treatment (21). The gut microbiome of children with SAM has distinct features from non-malnourished children from the same communities (22). These differences include lower gut microbiome bacterial diversity, as well as immature bacterial community compositions consistent with stunted/arrested microbiota development relative to children of the same age in the same environment without AM (22). Furthermore, nutritional rehabilitation with specially formulated prebiotic foods shifts the microbiome composition of acutely malnourished children toward their healthier counterparts throughout treatment (23). These microbiome shifts are accompanied by increases in the plasma proteome for mediators of bone growth and central nervous system development and are associated with the presence of beneficial bacteria enriched in pathways for carbohydrate metabolism (23).

Given the importance of the gut microbiome in nutritional status and evidence that microbiome composition is correlated with AM treatment response, we hypothesized that gut microbiome composition, including the potential metabolic pathways and ARGs carried by the bacteria within the microbiome at CMAM discharge among children recovered from uncomplicated SAM, may predict future relapse to AM. We also hypothesized that the gut microbiomes of children discharged following full anthropometric recovery may be different from those discharged with MAM. Our study used microbiome composition data from rectal swabs from a cohort of children treated for uncomplicated SAM and subsequently assessed for relapse to AM in South Sudan, a country with a high rate of AM relapse (5, 24). The aim of the study was to define the bacterial composition, as well as the ARGs and predicted metabolic pathways within the microbiome that may modify treatment response, that are associated with risk of

relapse to AM at 1 month or with differences between children discharged with full anthropometric recovery or with MAM post-treatment.

## MATERIALS AND METHODS

### Study design

As part of a larger longitudinal multi-country prospective cohort study (515 children in South Sudan), this nested study followed children aged 6–24 months for 1 month after discharge from treatment for uncomplicated SAM in Northern Bahr el Ghazal, South Sudan (5). Eligible study participants met the national CMAM protocol's recovery criteria (WHZ ≥ −2, MUAC ≥ 125 mm, and/or without edema based on treatment admission criteria), had not recently received inpatient care, and had no condition present likely to impair food intake. Additional details on the study's design, population, and sample size calculations are provided in a previously published protocol (24).

At the completion of each month of follow-up after initial SAM treatment, children were classified as either having relapsed to AM or having sustained recovery based on both WHZ and MUAC (or defaulted or died). Out of 515 children, 490 had a successful rectal swab, and 152 of these were selected for metagenomic sequencing. These 152 stools were selected as we wanted stools from pairs of children who either relapsed to AM or sustained recovery that could be matched 1:1 based on age, sex, and the criterion used for follow-up (MUAC vs WHZ) according to nutritional status at 3-month follow-up; however, we eventually decided to perform our analysis on the earlier 1-month timepoint as this was the closest timepoint to sample collection. Children who were discharged as having met national CMAM protocol recovery criteria (MUAC > 125 mm) but remained below the WHZ threshold for WHO recovery criteria (WHZ < −2) were separated into the "discharged with MAM" category for analysis (22 children). These 22 children (13 females and 9 males) were only included in analyses of microbiome features in the context of nutritional status at discharge and not for any analyses involving nutritional status at follow-up. Metadata for all participant demographics are included in Data S1.

This study was given approval for IRB exemption under IRB 202301022 by the Washington University in St. Louis School of Medicine's Human Research Protection Office.

### Data collection

Study enrollment took place at CMAM discharge, where written informed consent was obtained from each child's caregiver in the presence of a witness. Individual and household characteristics, including sex, age, and place of residence, were collected using a standardized enrollment questionnaire. CMAM treatment records were abstracted retrospectively to capture a child's treatment duration, medications received, and treatment response. Nutritional status was assessed by trained study staff through the measurement of height, weight, and edema at enrollment and again for 6 months of follow-up. At each monthly visit, children were classified as either having sustained recovery, relapsed to AM, died, or having missed a visit (unable to complete a visit within 3 weeks of the scheduled date). AM was defined as falling under either a WHZ < −2 SD, an MUAC < 125 mm, and/or the presence of bilateral nutritional edema. *Z*-scores for all anthropometric indices were determined first by study staff and then confirmed in Stata v17 using the zscore06 package, based on WHO 2006 Growth Standards (25, 26).

At CMAM discharge, rectal swabs were collected by trained nurses. Prior to collection, the process was explained to the child and caregiver, and consent was confirmed. Flocked nylon rectal swabs (eSwab, Copan Diagnostics, catalog #484CE) were obtained in a private space of the clinic in the presence of the caregiver and placed in tubes containing liquid amies solution to elute the sample. Approximately 40 µL of the eluted sample was applied per sampling spot onto Whatman FTA Elute cards (Qiagen, catalog

# WB120410), which were left to air dry before being sealed individually in plastic bags with silica gel sachets and humidity indicators for long-term storage. Discarded materials were placed in infectious waste bins, and disposal followed national guidelines.

Samples were transferred daily to a central field laboratory, recorded in both clinic and laboratory logbooks, and accompanied by transfer notes to verify counts. At the central field laboratory, samples were stored in a refrigerator to maintain stable temperatures, with desiccants and humidity indicators inspected and replaced monthly until they were shipped internationally for analysis to the LSHTM. International shipments between LSHTM and Washington University at St. Louis followed the same chain-of-custody procedures, with sample inventories prepared in advance, cross-checked upon receipt at the destination laboratory, and overseen by the research coordinator.

## DNA extraction, library preparation, and sequencing

Traces of stool were isolated from the Whatman cards by adding cards directly to bead tubes and resuspending in 1,000 µL CDI buffer from the DNeasy PowerSoil Pro Kit (Qiagen). DNA extraction was performed per manufacturer instructions with two rounds of bead beating for 2 min at 2,500 oscillations/min on a Mini-Beadbeater-24 (Biospect Products) with 5 min on ice in between. Nextera libraries were created with 0.5–1 ng input DNA with the Nextera flex reagents (Illumina) and purified using the Agencourt AMPure XP system (Beckman Coulter). DNA and sequencing library quantification was performed with the Quant-iT PicoGreen Fluorescence Assay (Invitrogen) and Qubit fluorometer dsDNA HS assay (Invitrogen). Pooled libraries were submitted for 2 × 150 paired-end sequencing on the Illumina NovaSeq 6000 platform. Stool samples were sequenced to a target read depth of 10 million paired reads per sample. Read counts for each sample are provided in Data S3. Samples with less than 100,000 post-processed reads were excluded from downstream analyses. All 152 stool samples passed this quality control metric.

## Computational sequencing and read processing

All metagenomic reads were processed using FastQC v0.12.1 (27). Sequencing adapters were trimmed using Trimmomatic v0.39 with parameters "ILLUMINACLIP:NexteraPE-PE.fa:2:30:10:1:TRUE LEADING:10 TRAILING:10 SLIDINGWINDOW:4:15 MINLEN:60" (28). Human reads were removed with Deconseq v0.4.3 with reference to human chromosomes (29). Disordered reads or reads with eliminated mates were re-paired with the repair.sh script in BBMap v38.63 (30). Post-processing read counts are provided in Data S3.

## Analysis of microbial composition, ARGs, and functional potential

To identify microbial composition, metagenomic reads were used as input to MetaPhlAn4 v4.0.2 using the vOct22 database (31). Species calls from MetaPhlAn4 were used to calculate the richness and Shannon diversity of metagenomic samples using the vegan (version 2.6-8) package in R. To identify ARG content, metagenomic reads were used as input to ShortBRED v0.9.5 using the Comprehensive Antibiotic Resistance Database version 3.2.6 and NCBI's AMRFinderPlus database version 2023-02-23.1 (32–34). Shortbred_quantify.sh with default settings was used to identify the abundance and identity of antimicrobial resistance gene loci from metagenomic reads to markers from the aforementioned databases. To identify microbial functional potential, processed metagenomic reads were used as input to HUMAnN3 (35) v3.6 using the UniRef90 v201901 database and mapped against the Kyoto Encyclopedia of Genes and Genomes (KEGG) Orthogroups and Gene Ontology (GO) databases. humann, humann_renorm, humann_regroup, and humann_rename were used with default settings to identify the abundance of gene families associated with functional KEGG and GO categories. Differences in richness, Shannon diversity, abundance of ARGs, unique number of ARGs, abundance of beta-lactam ARGs, unique number of beta-lactam ARGs, ratio of

beta-lactam ARGs to all ARGs, and unique number of KEGG and GO pathways were compared for a number of categorical and continuous variables of nutritional states (categorical: nutritional status at discharge, relapse to AM at 1-month follow-up, sex, continuous: WHZ, MUAC, height-for-age *z*-score (HAZ), weight-for-age *z*-score (WAZ) at 1-month follow-up, change in WAZ, WHZ, HAZ, MUAC from discharge to 1-month follow-up, age). Beta-lactam ARGs were identified as any ARG that was flagged with "BETA LACTAM" in the "class" column of our ShortBRED output. Binary occurrences of all ARGs, including beta-lactam ARGs, were computed by counting each unique entry in the "Family" column as a unique ARG hit. Proportions of total or unique beta-lactam gene content to all ARG genes were calculated by dividing the sum of reads per kilobase per million mapped reads (RPKM) devoted to beta-lactam ARGs by the RPKM of all ARGs, or the counts of unique beta-lactam ARGs by the counts of all unique ARGs. Proportions in the text are reported as decimal values and can be converted to percentages by multiplying values by 100. Analysis was performed using Mann-Whitney and Kruskal-Wallis in R Studio to analyze relationships of microbiome features with categorical variables. Linear regression using the lm() function from lme4 1.1.35.5 in R Studio was used to analyze relationships of microbiome features with continuous variables. The Benjamini-Hochberg (BH) correction was used to adjust *P* values for multiple hypotheses when applicable. Distribution of microbiome features such as species calls, ARG profiles, and functional potential profiles was used to calculate Bray-Curtis dissimilarity between different groups (nutritional status at discharge, relapse to AM at 1-month follow-up). Bray-Curtis dissimilarities were used for principal coordinate analysis (PCoA) and visualization of sample pair-wise dissimilarity distributions. PERMANOVA (adonis2 and vegan) in R Studio was utilized to analyze Bray-Curtis dissimilarity. Overall carriage of ARGs was measured in RPKM, and binary carriage of ARGs is measured in counts. Reports of ARG content in the text are based on median values. Children with missing amoxicillin exposure records (in_amx = NA) were excluded from any analyses involving amoxicillin exposure; this is different from amoxicillin. Binary carriage of KEGG and GO pathways was measured in counts. Reports of microbiome feature values in the text are based on median values.

## Metadata associations with specific microbiome features

Metadata variables in Data S1 associated with the microbiome were completed on the following variables of interest: MUAC at discharge (muac0), WHZ at discharge (whz0), WAZ at discharge (waz0), HAZ at discharge (haz0), nutritional status at discharge (fullrecovery0), MUAC at 1-month follow-up (muac1), WHZ at 1-month follow-up (whz1), WAZ at 1-month follow-up (waz1), HAZ at 1-month follow-up (haz1), relapse to AM at 1-month follow-up defined by MUAC or WHZ with both indicators used for defining recovery at discharge (relapse1am_whz_and_muac), change in MUAC between discharge and 1-month follow-up (muacchange0to1), change in WHZ between discharge and 1-month follow-up (whzchange0to1), change in WAZ between discharge and 1-month follow-up (wazchange0to1), change in HAZ between discharge and 1-month follow-up (hazchange0to1), change in weight between discharge and 1-month follow-up (weightchange0to1), change in height between discharge and 1-month follow-up (heightchange0to1), sex (female), and age of child at stool collection (age0). MaAsLin2 1.18.0 was used with default parameters, including default feature prevalence filters, to determine associations. Taxonomy (MetaPhlAn4), antimicrobial resistance gene carriage (shortBRED), and microbiome functional potential (HUMAnN3) were used as input data/associations of interest for separate MaAsLin2 commands. Only features that had both *P* values< 0.05 and *q* values< 0.05 (BH adjusted values) were declared significant.

## RESULTS

### Demographics

A total of 515 children diagnosed with uncomplicated SAM were recruited upon admission to treatment at six clinics between April 2021 and June 2022 and were followed throughout their CMAM treatment course in South Sudan (5). Children who achieved nutritional recovery were then also followed for an additional month post-discharge. A total of 490 of these children had a successful rectal swab collection, and a subset of 152 rectal swabs were processed for metagenomic sequencing (Fig. 1A). All rectal swabs from 152 children were successfully sequenced. Of these 152 children, 130 children fully met the 2023 WHO criteria of recovery at discharge (referred to as discharged recovered or discharged without AM), and 22 children met South Sudan CMAM discharge criteria, but not the WHO definition of recovery at discharge due to their WHZ remaining below −2 (referred to as discharged with MAM) (3, 4). The discrepancy in the two criteria is due to WHO standards requiring meeting both WHZ and MUAC cutoffs, while exit from CMAM in South Sudan was based on achieving a healthy measurement in the same anthropometric index with which the child was admitted (either WHZ or MUAC) (4). The 22 children who were discharged with MAM were used to investigate microbiome differences that may exist between children discharged recovered or with MAM but were excluded from any analyses involving 1-month relapse follow-up data to avoid confounding variables that may be introduced by including children who were discharged already still acutely mal-nourished. In addition to these relapse statuses, we also used anthropometric measure-ments as continuous variables. This was to account for the fact that AM recovery in CMAM treatment is definitional rather than biological and to evaluate anthropometric trajectories as a proxy for nutritional status or risk factor for AM relapse outside of these definitional categories. Approximately 70% of participants were female, with a mean age of 14.5 months at enrollment. From 61% to 75% of children had documented exposure to amoxicillin; there were no differences in documented amoxicillin exposures between children who sustained recovery or those who relapsed to AM at 1-month follow-up ($P > 0.05$; Table 1). CMAM protocol indicates that all children should receive amoxicillin during treatment. Children who had no amoxicillin exposure in their documentation may have received amoxicillin without nurses noting the encounter on their health records, or breaks in the amoxicillin supply chain in South Sudan may have resulted in actual lapses in amoxicillin administration (4). The prevalence of relapse to AM at 1-month post-discharge in children discharged fully recovered with both anthropometric indices was 30.0% of which 6.2% represented relapse to SAM. Children whose recovery was sustained for at least 1 month after discharge had higher anthropometric indices at discharge than those who relapsed ($P < 0.0001$; Table 1) (5).

### Overall gut microbiome composition at initial SAM recovery is not predictive of subsequent relapse to AM at 1-month post-discharge

The gut microbiome richness and Shannon diversity of children at initial SAM recovery were not significantly different between those who reached full anthropometric recovery (i.e., fully recovered, richness = 111, and Shannon = 3.2) and those who were discharged with MAM (richness = 105 and Shannon = 3.3; Fig. S1A and B). Gut microbiome richness and Shannon diversity at discharge were also not significantly different between children who subsequently relapsed to AM (richness = 110 and Shannon = 3.2) or sustained recovery (richness = 111 and Shannon = 3.2) at the 1-month follow-up timepoint (Fig. 1B and C). The length of time spent in CMAM treatment did not have an effect on richness or Shannon diversity in our cohort ($r^2 = -0.0061$ and $r^2 = -0.0056$, respectively; Fig. S1C and D).

Sex and age were associated with overall microbiome differences of children in our cohort, with female sex associated with greater species richness (female = 116, male = 91, $P = 0.0026$) but not greater Shannon diversity (female: 3.3 and male = 3.1), as has

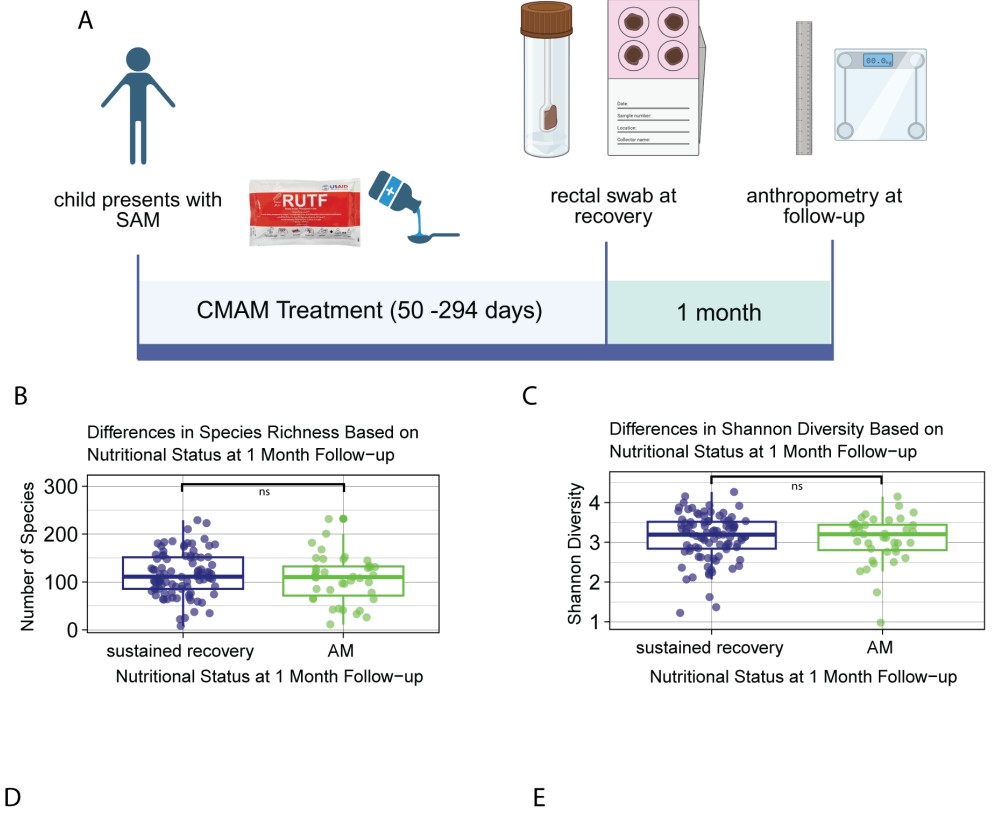

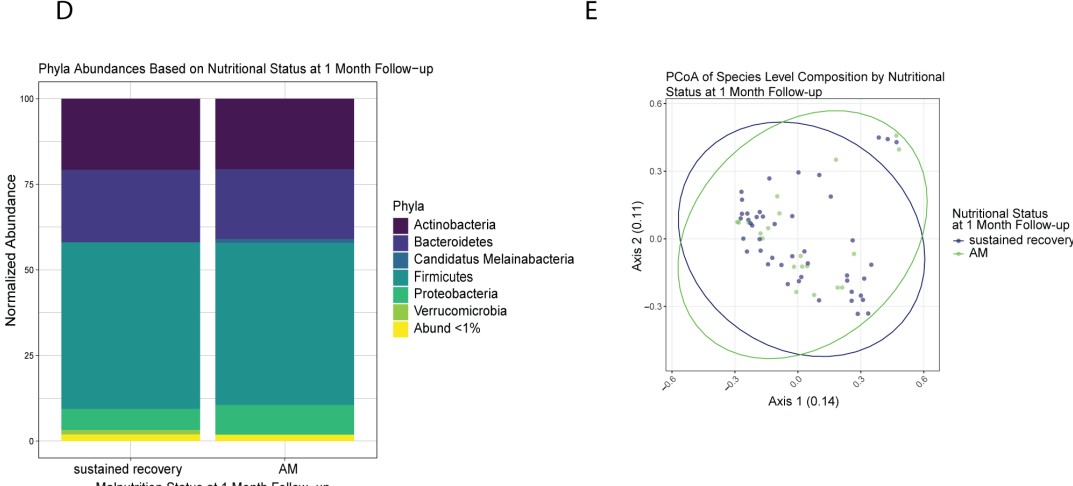

FIG 1 Broad gut microbial composition associations with relapse to acute malnutrition. (A) Study workflow and collection, rectal swabs were collected from children immediately following discharge from CMAM for SAM. Children were followed 1-month post-discharge to assess relapse to acute malnutrition. Created in BioRender. https://BioRender.com/bdxedn4. (B) Species richness in the gut microbiomes at discharge for children who sustained recovery or relapsed to acute malnutrition at 1-month follow-up. Mann-Whitney with BH correction (relapse: 111, sustained recovery: 110, $P = 0.41$). (C) Shannon diversity in the microbiomes at discharge for children who sustain recovery or relapse to acute malnutrition at 1-month follow-up. Mann-Whitney with BH correction (relapse: 3.2, sustained recovery: 3.2, $P = 0.73$). (D) Normalized abundance of phyla distribution in the microbiomes at discharge of children who sustained recovery or relapsed to acute malnutrition at 1-month follow-up. MaAsLin2 (Firmicutes $P = 0.059$, Proteobacteria $P = 0.43$, Bacteroidetes $P = 0.37$, Candidatus Melainabacteria $P = 0.26$, Verrucomicrobia $P = 0.45$, and Actinobacteria $P = 0.90$). (E) PCoA of species distribution in the microbiomes at discharge of children who sustain recovery or relapse to acute malnutrition at 1-month follow-up. Bray-Curtis dissimilarity PERMANOVA ($F = 1.3$, Pr [>$F$] = 0.16). The two PCoA axes explain 14% and 11% of the variation in microbiome species distribution among samples, respectively.

been documented in other studies (36, 37). Increasing age was associated with both greater species richness and Shannon diversity (richness: $r^2 = 0.14$, $P = 1.1e-06$, Shannon: $r^2 = 0.033$, $P = 0.014$; Fig. S2A through D). To account for these age-related effects, we

**TABLE 1** Demographics of study participants[a,b,c]

| Characteristic | Total sample N = 152 | Met WHO recovery criteria at discharge N = 130 | Month 1 post-discharge | | |
|---|---|---|---|---|---|
| | | | Sustained recovery N = 91 | Relapsed to AM N = 39 | P-value |
| Sex and age | | | | | |
| Female | 105 (69.1%) | 92 (70.8%) | 67 (73.6%) | 25 (64.1%) | 0.274 |
| Age at enrollment (months) | 14.5 (3.8) | 14.4 (3.8) | 14.5 (3.8) | 14.0 (3.7) | 0.225 |
| Initial treatment characteristics | | | | | |
| MUAC at discharge (mm) | 129 (3.9) | 129 (4.1) | 130 (4.3) | 128 (3.1) | 0.010 |
| WHZ at discharge | −1.2 (0.9) | −1.1 (0.8) | −0.95 (0.8) | −1.3 (0.7) | 0.010 |
| WAZ at discharge | −2.1 (0.8) | −2.0 (0.8) | −1.9 (0.8) | −2.2 (0.7) | 0.015 |
| WAZ ≥ −3 to < −2 | 53 (34.2%) | 43 (33.1%) | 25 (27.5%) | 18 (46.2%) | 0.038 |
| WAZ < −3 | 23 (15.1%) | 14 (10.8%) | 9 (9.9%) | 5 (12.8%) | 0.621 |
| HAZ at discharge | −2.3 (1.4) | −2.3 (1.4) | −2.2 (1.4) | −2.4 (1.4) | 0.302 |
| HAZ ≥ −3 to < −2 | 34 (22.4%) | 28 (21.5%) | 20 (22.0%) | 8 (20.5%) | 0.852 |
| HAZ < −3 | 48 (31.6%) | 42 (32.3%) | 27 (29.7%) | 15 (38.5%) | 0.326 |
| Received documented course of amoxicillin during treatment | 100 (65.8%) | 85 (65.4%) | 56 (61.5%) | 29 (74.4%) | 0.221 |
| Post-discharge anthropometry | | | | | |
| MUAC at 1-month post-discharge (mm) | 129 (7.1) | 129 (7.4) | 132 (6.0) | 123 (6.6) | <0.0001 |
| MUAC ≥ 115 to < 125 | 22 (14.5%) | 18 (13.9%) | 0 (0%) | 18 (46.2%) | <0.0001 |
| MUAC <115 | 5 (3.3%) | 5 (3.9%) | 0 (0%) | 5 (12.8%) | <0.0001 |
| Average change in MUAC, discharge to 1-month post-discharge | −0.23 (5.2) | −0.3 (5.2) | 1.8 (3.7) | −5.2 (5.0) | <0.0001 |
| WHZ at 1-month post-discharge | 1.5 (1.1) | −1.4 (1.1) | −1.0 (0.9) | −2.3 (0.9) | <0.0001 |
| WHZ ≥ −3 to < −2 | 38 (25.0%) | 25 (19.2%) | 0 (0%) | 25 (64.1%) | <0.0001 |
| WHZ < −3 | 7 (4.6%) | 5 (3.4%) | 0 (0%) | 5 (12.8%) | 0.0005 |
| Average change in WHZ, discharge to 1-month post-discharge | −0.2 (0.9) | −0.3 (0.8) | −0.02 (0.7) | −1.0 (0.9) | <0.0001 |
| WAZ at 1-month post-discharge | −2.3 (1.0) | −2.2 (0.9) | −1.9 (0.9) | −2.9 (0.8) | <0.0001 |
| WAZ ≥ −3 to < −2 | 57 (37.5%) | 49 (37.7%) | 31 (34.1%) | 18 (46.2%) | 0.193 |
| WAZ < −3 | 34 (22.4%) | 24 (18.5%) | 8 (8.8%) | 16 (41.0%) | <0.0001 |
| Average change in WAZ, discharge to 1-month post-discharge | −0.2 (0.6) | −0.2 (0.6) | 0.001 (0.5) | −0.7 (0.6) | <0.0001 |
| HAZ at 1-month post-discharge | −2.3 (1.4) | −2.3 (1.4) | −2.3 (1.4) | −2.4 (1.3) | 0.326 |
| HAZ ≥ −3 to < −2 | 38 (25.0%) | 34 (26.2%) | 23 (25.3%) | 11 (28.2%) | 0.728 |
| HAZ < −3 | 48 (31.6%) | 39 (30.0%) | 26 (28.6%) | 13 (33.3%) | 0.587 |
| Average change in HAZ, discharge to 1-month post-discharge | −0.02 (0.3) | −0.003 (0.3) | −0.01 (0.3) | 0.01 (0.4) | 0.625 |
| Average change in weight (kg), discharge to 1-month post-discharge | 0.04 (0.6) | 0.003 (0.6) | 0.2 (0.5) | −0.4 (0.5) | <0.0001 |
| Average change in height (cm), discharge to 1-month post-discharge | 0.9 (0.9) | 0.9 (0.9) | 0.9 (0.9) | 1.0 (0.9) | 0.635 |

[a]AM, acute malnutrition; MUAC, mid-upper arm circumference; WHZ, weight-for-height z-score; WAZ, weight-for-age z-score; HAZ, height-for-age z-score.
[b]Frequency and percent for binary variables, mean, and standard deviation for continuous variables. P values determined between sustained recovery and relapse to AM.
[c]Demographics data of study participants. Percentage of binary variables reported in brackets next to frequency. Standard deviation of continuous variables is reported in brackets next to the mean. P values correspond to differences between sustained recovery and relapse to AM.

next looked at microbiome richness and Shannon diversity in associations with age-stratified anthropometric changes. No significant associations (all $P > 0.05$) were found between higher measurements 1-month post-discharge in WHZ, WAZ, and HAZ with Shannon diversity or richness (Fig. S3). Higher MUAC at 1-month post-discharge in children 1 year or older was not associated with changes in Shannon diversity or richness, and higher MUAC in children younger than 1 year of age was correlated with small increases in Shannon diversity (adjusted $r^2$: 0.072, $P = 0.037$) but not richness (Fig.

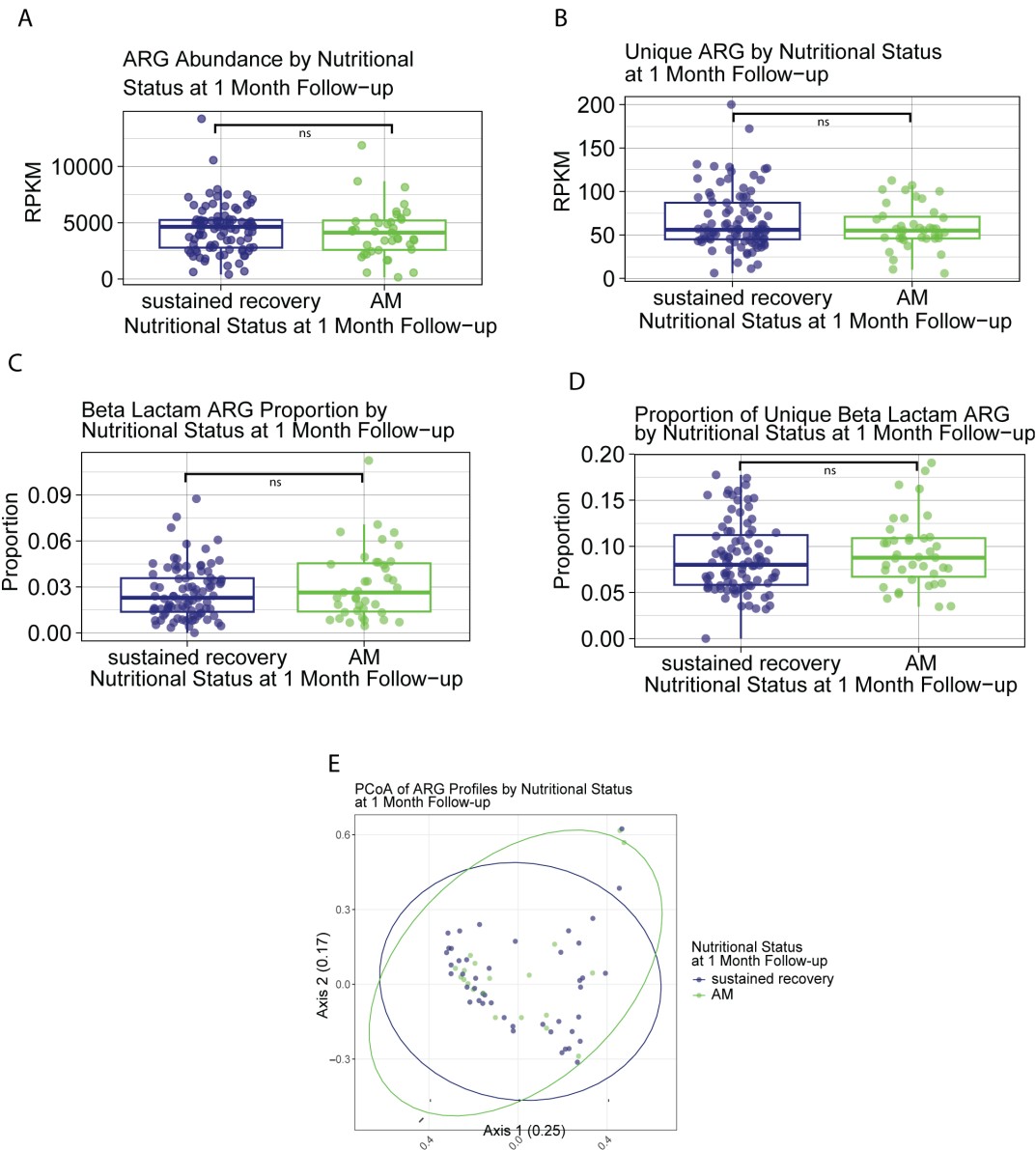

**FIG 2** Broad microbiome antimicrobial resistance gene associations with relapse to acute malnutrition. (A) ARG abundance in RPKM in the microbiomes at discharge of children who sustained recovery or relapsed to acute malnutrition at 1-month follow-up. Mann-Whitney with BH correction (relapse: 4,122, sustain recovery: 4,636, $P = 0.36$). (B) Unique ARG hits in counts in the microbiomes at discharge of children who sustained recovery or relapsed to acute malnutrition at 1-month follow-up. Mann-Whitney with BH correction (relapse: 55, sustain recovery: 56, $P = 0.58$). (C) Proportion of gene content devoted to beta-lactam ARGs in the microbiomes at discharge of children who sustained recovery or relapsed to acute malnutrition at 1-month follow-up. Mann-Whitney with BH correction (relapse: 0.026, sustain recovery: 0.023, $P = 0.27$). (D) Proportion of unique genes that encode beta-lactam resistance in the microbiomes at discharge of children who sustained recovery or relapsed to acute malnutrition (relapse: 0.08772, sustain recovery: 0.080, $P = 0.4$). (E) PCoA of ARG distribution in the microbiomes at discharge of children who sustained recovery or relapsed to acute malnutrition at 1-month follow-up. Bray-Curtis dissimilarity PERMANOVA ($F = 0.82$, Pr [$>F$] = 0.57). The two PCoA axes explain 25% and 17% of the variation in microbiome ARG distribution among samples, respectively.

S3B and F). Positive changes in WHZ, WAZ, and HAZ between discharge and 1-month follow-up were also not associated with richness or Shannon diversity (all $P > 0.05$; Fig. S4). Positive changes in MUAC between discharge and 1-month follow-up in children 1 year or older were not associated with Shannon diversity or richness, and positive changes in MUAC between discharge and 1-month follow-up in children younger than 1 year of age at sampling were correlated with small increases in Shannon diversity (adjusted $r^2$: 0.066, $P = 0.044$) but not richness (Fig. S4B and F).

The average abundance of phyla within the gut microbiome at discharge of children who fully recovered was not significantly different between those who sustained recovery and those who relapsed to AM at 1-month follow-up (Fig. 1D). The main phyla that dominated these children's microbiota were Firmicutes (sustained recovery: 50.2% and relapsed to AM: 49.1%), Bacteroidetes (sustained recovery: 21.8% and relapsed to AM: 21.2%), and Actinobacteria (sustained recovery: 21.5% and relapsed to AM: 21.4%; Fig. 1D). The average abundance of phyla among children discharged with MAM was not significantly different from those discharged fully recovered (Fig. S5A). Children who were discharged with MAM also had microbiomes dominated by Firmicutes (49.3%), Bacteroidetes (21.2%), and Actinobacteria (18.9%; Fig. S5A). We did not observe any overall differences in the relative abundance of the bacterial phyla present in the microbiome at discharge of children who sustained recovery and those who relapsed to AM at the 1-month follow-up with MaAsLin2 (Fig. 1D). Similarly, we did not observe any differences in the distribution of species determined by PCoA with Bray-Curtis dissimilarity within the microbiome communities of those who sustained recovery and those who relapsed to AM at one month follow-up ($P > 0.05$, Fig. 1E). These lack of differences at discharge also applied to the relative abundance of bacterial phyla and the species distribution profiles of the microbiomes of children who were discharged fully recovered or with MAM (Fig. S5A and B).

Overall, the microbiota at discharge of children discharged from CMAM treatment for uncomplicated SAM in this population was largely unrelated to anthropometry changes, whether a child was discharged fully recovered or with MAM, or their risks of AM relapse 1-month post recovery from AM.

## Overall antibiotic resistance gene distribution at initial SAM recovery is not associated with subsequent relapse to AM at 1-month post-discharge

Due to the administration of antibiotics, specifically amoxicillin, as a routine part of CMAM treatment for SAM in South Sudan and in our cohort (Table 1), we sought to investigate the relationship between ARGs and nutritional status. We hypothesized that the presence and distribution of ARGs in the microbiome might indicate increased antibiotic resistance and decreased efficacy of antibiotic treatment during CMAM. These changes may influence the degree of recovery following CMAM, such as differences between full recovery and recovery to MAM, and may predict the likelihood of AM relapse. Neither overall carriage of ARGs nor the number of unique ARG hits were significantly different in the microbiomes between children who relapsed to AM (total: 4,122 and unique: 55) and those with sustained recovery (total: 4,636 and unique: 56) at 1-month post-discharge (Fig. 2A and B). Overall and unique microbiome ARG content also did not discriminate between children who were discharged as fully recovered (total: 4,369 and unique: 56) or those discharged as partially recovered with MAM (total: 2,884 and unique: 61; Fig. S6A and B). Length of CMAM treatment was not significantly associated with changes in total or unique ARG hits ($r^2 = -0.0059, -0.0060$, respectively; Fig. S6C and D). Documented amoxicillin exposure was not correlated with differences in total (Fig. S7A and B; exposed: 4,120 and non-exposed: 4,650) or unique ARGs (exposed: 57 and non-exposed: 56), with the caveat that the details of antibiotic administration, including timing relative to sample collection and antibiotic compliance, were poorly documented in this cohort.

Due to the common usage of beta-lactam antibiotics such as amoxicillin in the CMAM model of treating AM, we specifically looked at the changes in beta-lactam ARGs in the gut microbiome (4, 38). The total number of beta-lactam ARGs in the microbiome was not significantly different between children who sustained recovery at 1-month follow-up (94) and those who relapsed to AM (88) (Fig. S8A). The number of unique beta-lactam ARGs in the microbiome also did not differ between children who sustained recovery (5) and those who relapsed to AM (5) (Fig. S8B). Similarly, the total number of beta-lactam ARGs in the microbiome did not differ for children discharged fully recovered (93) and those discharged with MAM (95) (Fig. S8C), nor did it differ for the

unique number of beta-lactam ARGs in the microbiome between children discharged fully recovered (5) and those discharged with MAM (5) (Fig. S8D). Length of CMAM treatment did not have an effect on the total number of beta-lactam ARGs nor the number of unique beta-lactam ARGs ($r^2 = -0.0017$ and $-0.0069$, respectively; Fig. S8E and F). We did not observe a difference in the gene content devoted to beta-lactam ARGs in the gut microbiome, defined as the ratios of all beta-lactam ARGs to all ARGs, between children who sustained recovery at 1-month follow-up (0.023) and those who relapsed to AM (0.026; Fig. 2C). Gene content devoted to beta-lactam ARGs was also not different between children discharged fully recovered (0.023) and those discharged with MAM (0.030; Fig. S9A). Ratios of all unique beta-lactam ARGs to unique ARGs did not differ between those who sustained recovery (0.080) and those who relapsed to AM (0.088; Fig. 2D). The ratios of all unique beta-lactam ARGs to unique ARGs also did not differ between those who were discharged fully recovered (0.083) and those who were discharged with MAM (0.084; Fig. S9B). Length of CMAM treatment did not have an effect on the gene content devoted to beta-lactam ARGs nor the ratios of all unique beta-lactam ARGs to all unique ARGs ($r^2 = 0.00071$ and $-0.0054$, respectively; Fig. S9A and B). The total number of beta-lactam ARGs was not affected by whether or not a child had a documented amoxicillin exposure (exposed: 87 and non-exposed: 123; Fig. S10A). Similarly, the number of unique beta-lactam ARGs was also not affected by documented amoxicillin exposure (exposed: 5 and non-exposed: 6; Fig. S10B). The gene content devoted to beta-lactam ARGs in the microbiome was not affected by documented amoxicillin exposure (exposed: 0.026 and non-exposed: 0.027; Fig. S10C). Likewise, the ratios of unique beta-lactam ARGs to unique ARGs were also not affected by documented amoxicillin exposure (exposed: 0.087 and non-exposed: 0.080; Fig. S10D).

The overall profiles of ARGs in the microbiome at discharge determined by PCoA with Bray-Curtis dissimilarity did not significantly differ between children who sustained recovery or those who relapsed to AM at 1-month follow-up (Fig. 2E). However, this overall profile did differ between children discharged recovered and those discharged with MAM ($F = 2.3$, Pr [>$F$] = 0.022; Fig. S11). This suggests that meeting the WHO standards for AM recovery with CMAM treatment may modulate the ARGs present in the microbiome, that children discharged with MAM may not have received sufficient antibiotics to achieve the same distribution of ARGs in their microbiome, or that differences in ARG profiles may result in decreased responses to CMAM treatment.

Sex and age were not discriminatory for any changes in the number of total gut microbiome ARGs at discharge (male: 4,517, female: 4,122, age: $r^2 = 0.0085$; Fig. S12A and 13A). Sex and age were also not discriminatory for any changes in unique gut microbiome ARGs at discharge (male: 58, female: 55, age: $r^2 = -0.0063$; Fig. S12B and 13B). The number of total beta-lactam ARGs in the gut microbiome at discharge was not affected by sex or age (male: 86, female: 98, age: $r^2 = 0.014$; Fig. S12C and 13C). Similarly, the number of unique beta-lactam ARGs in the gut microbiome at discharge was also unaffected by sex or age (male: 5, female: 5, age: $r^2 = -0.0060$; Fig. S12D and 13D). Sex was not associated with any changes in the proportion of total beta-lactam gut microbiome ARGs to all ARGs (male: 0.022 and female: 0.025; Fig. S12E), nor was it associated with the proportion of unique beta-lactam ARGs to all unique ARGs present in the gut microbiome (male: 0.079 and female: 0.087; Fig. S12F). Age was associated with increases in proportion of total beta-lactam ARGs to all ARGs in the gut microbiome ($r^2 = 0.057$, $P = 0.0018$; Fig. S13E); however, it was not associated with proportion of unique beta-lactam ARGs to all unique ARGs ($r^2 = -0.0064$; Fig. S13F).

Overall, these data suggest that ARGs within the microbiome of children at discharge in our cohort are largely unrelated to their risks of relapse to AM but are related to whether or not they were discharged fully recovered. The distribution of ARGs may be affected by age at sampling and whether a child fully meets the WHO's anthropometric criteria for recovery following CMAM treatment.

## Overall microbial functional potential at initial SAM recovery does not predict subsequent relapse to acute malnutrition at 1-month post-discharge

The number of unique predicted GO and KEGG pathways present in the microbiota of children treated for SAM at discharge did not differ between those who were discharged with MAM (GO: 3,386 and KEGG: 2,760) and those who were fully recovered (GO: 3,066 and KEGG: 2,631; Fig. S14A and B), nor were they different between children who relapsed to AM (GO: 3,157 and KEGG: 2,664) or sustained recovery (GO: 3,033 and KEGG: 2,626) at 1-month follow-up (Fig. 3A and B). The length of time spent in CMAM treatment did not have an effect on the number of unique KEGG and GO pathways present in the microbiome of children in our cohort ($r^2 = -0.0066$ and $-0.0071$, respectively; Fig. S14C and D).

Similar to ARGs, sex (male: GO: 3,251 and KEGG: 2,720; female: GO: 3,036 and KEGG: 2,638) and age (GO: $r^2 = -0.0057$ and KEGG: $r^2 = -0.0032$) were not drivers for any differences in the number of predicted GO and KEGG pathways (Fig. S15). PCoA with Bray-Curtis dissimilarity of GO and KEGG pathways was analyzed to compare the distribution of unique pathways by nutritional status. These distributions were not different between children discharged with MAM or fully recovered (Fig. S16), nor between children who sustained recovery or relapsed to AM at 1-month follow-up (Fig. 3C and D).

Overall, we found that predicted microbial functional potential within the gut microbiomes of children at discharge was not influenced by participant factors like sex or age, nor was it distinct between children with different nutritional status at discharge or 1-month follow-up.

## Specific microbiome features following treatment for SAM are associated with anthropometric differences

Given the lack of overall differences in bacterial composition, ARGs, and functional potential within the gut microbiomes of children who did or did not relapse to malnutrition at 1-month follow-up, we applied generalized linear mixed effect models with MaAsLin2 (39) to identify specific bacteria, ARGs, or potential gene pathways that may be correlated with these differences.

We did not identify any specific features within the gut microbiome of children at discharge that were associated with whether a child met the WHO standards of recovery at discharge, nor whether a child sustained recovery or relapsed to AM at 1-month follow-up. However, specific features were identified that differentiated children with WHZ or MUAC measurements at discharge, 1-month follow-up, or change in these measurements between those intervals. We could not attribute these features to any specific species with confidence. Children with positive WHZ change between discharge and 1-month follow-up had decreased potential for oxoglutarate synthase activity (coeff: $-12$, $q = 0.046$; Fig. 4A; Fig. S17A). Higher MUAC at 1-month follow-up was associated with increased trimethoprim-resistant dihydrofolate reductase ARGs (coeff: 2.1, $q = 0.0081$; Fig. 4B; Fig. S17B). Higher MUAC at 1-month follow-up was also associated with increased potential for alkylglycerone phosphate synthase (coeff: 1.5, $q = 0.040$), and menaquinone biosynthetic activity (coeff: 0.39, $q = 0.050$; Fig. 4C; Fig. S17C and D). We found positive changes in MUAC between discharge and 1-month follow-up were negatively correlated with increased potential for menaquinone biosynthetic activity (coeff: $-0.38$, $q = 0.032$; Fig. 4D; Fig. S17E). Higher WHZ at discharge was associated with increased relative abundance of *Sutterella wadsworthensis* (coeff: 6.6, $q = 0.041$; Fig. 4E; Fig. S17F). Higher MUAC at discharge was associated with increased trimethoprim-resistant dihydrofolate reductase ARGs (coeff: 0.90, $q = 0.014$; Fig. 4F; Fig. S17G). Higher MUAC at discharge was also associated with increased relative abundance of *Porphyromonas gingivalis* (coeff: 1.4, $q = 0.025$; Fig. 4G; Fig. S17H). Unlike WHZ and MUAC, WAZ and HAZ are not used for defining malnutrition; however, specific microbiome feature differences were also identified for changes in WAZ, and HAZ between discharge and 1-month follow-up (Data S2).

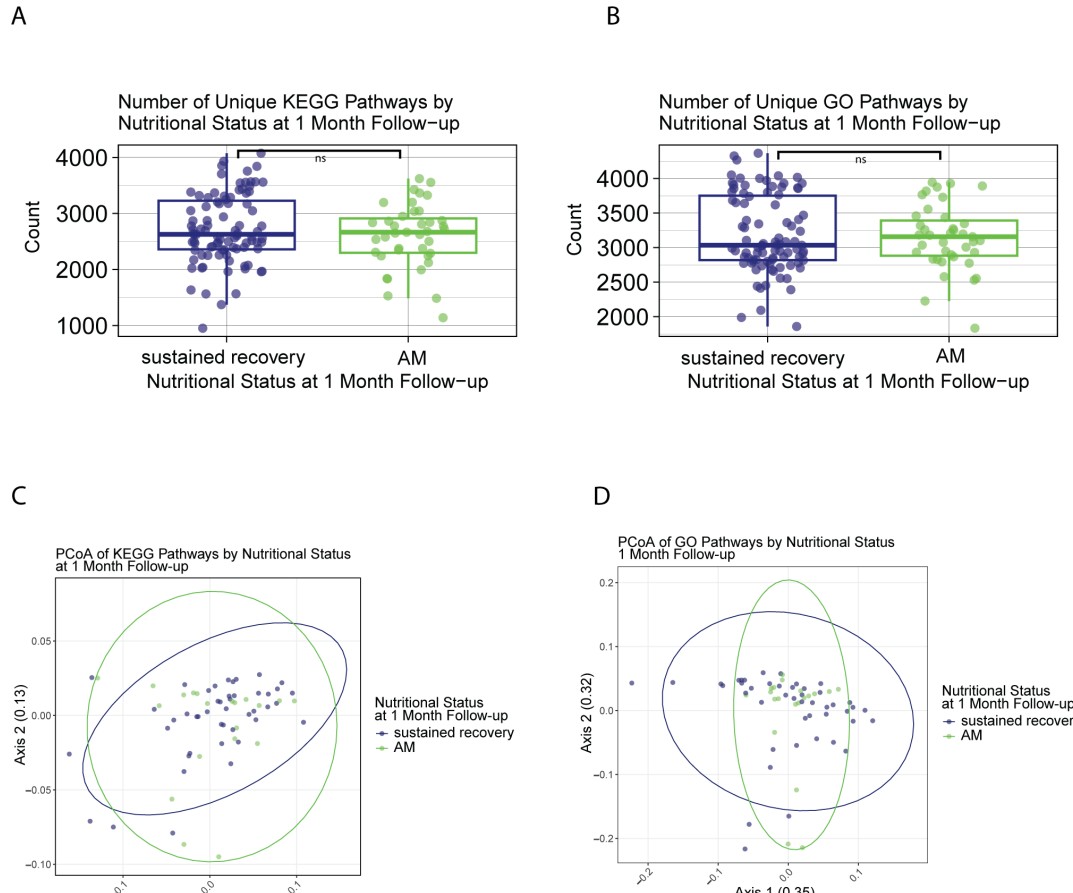

**FIG 3** Broad microbiome functional potential associations with relapse to malnutrition. (A) Unique KEGG pathways in counts in the gut microbiomes at discharge of children who sustained recovery or relapsed to acute malnutrition at 1-month follow-up. Mann-Whitney with BH correction (relapse: 2,664, sustain recovery: 2,626, $P = 0.55$). (B) Unique GO pathways in counts in the microbiomes at discharge of children who sustained recovery or relapsed to acute malnutrition at 1-month follow-up. Mann-Whitney with BH correction (relapse: 3,157, sustain recovery: 3,033, $P = 0.95$). (C) PCoA of KEGG pathway distribution in the microbiomes at discharge of children who sustained recovery or relapsed to acute malnutrition at 1-month follow-up. Bray-Curtis dissimilarity PERMANOVA ($F = 0.8713$, Pr [>$F$] = 0.362). The two PCoA axes explain 73% and 13% of the variation in microbiome functional potential defined by KEGG among samples, respectively. (D) PCoA of GO pathway distribution in the microbiomes at discharge of children who sustained recovery or relapsed to acute malnutrition at 1-month follow-up. Bray-Curtis dissimilarity PERMANOVA ($F = 0.4507$, Pr [>$F$] = 0.80). The two PCoA axes explain 35% and 32% of the variation in microbiome functional potential defined by GO among samples, respectively.

Anthropometry measurements are the clinical standard for defining AM (3, 40). Our results suggest that specific changes in the microbiome of children at discharge, such as increases in *Sutterella wadsworthensis* and trimethoprim-resistant dihydrofolate reductase ARGs, may be associated with differences in anthropometry that may be more sensitive to microbiome changes and indicate future health trajectories.

## DISCUSSION

In this study, we found that there were no overall differences in microbial composition, ARGs, or functional potential within the microbiomes of children discharged following treatment for SAM in South Sudan that can predict whether a child will relapse to AM at 1-month follow-up. These features also could not distinguish between children of different WHZ, MUAC, HAZ, or WAZ measurements at 1-month follow-up, including children with positive changes in those measurements between discharge and follow-up. Only higher MUAC measurements at 1-month follow-up and positive changes in MUAC between discharge and follow-up in children under 1 year of age were significantly associated with increasing Shannon diversity. Overall microbiome differences

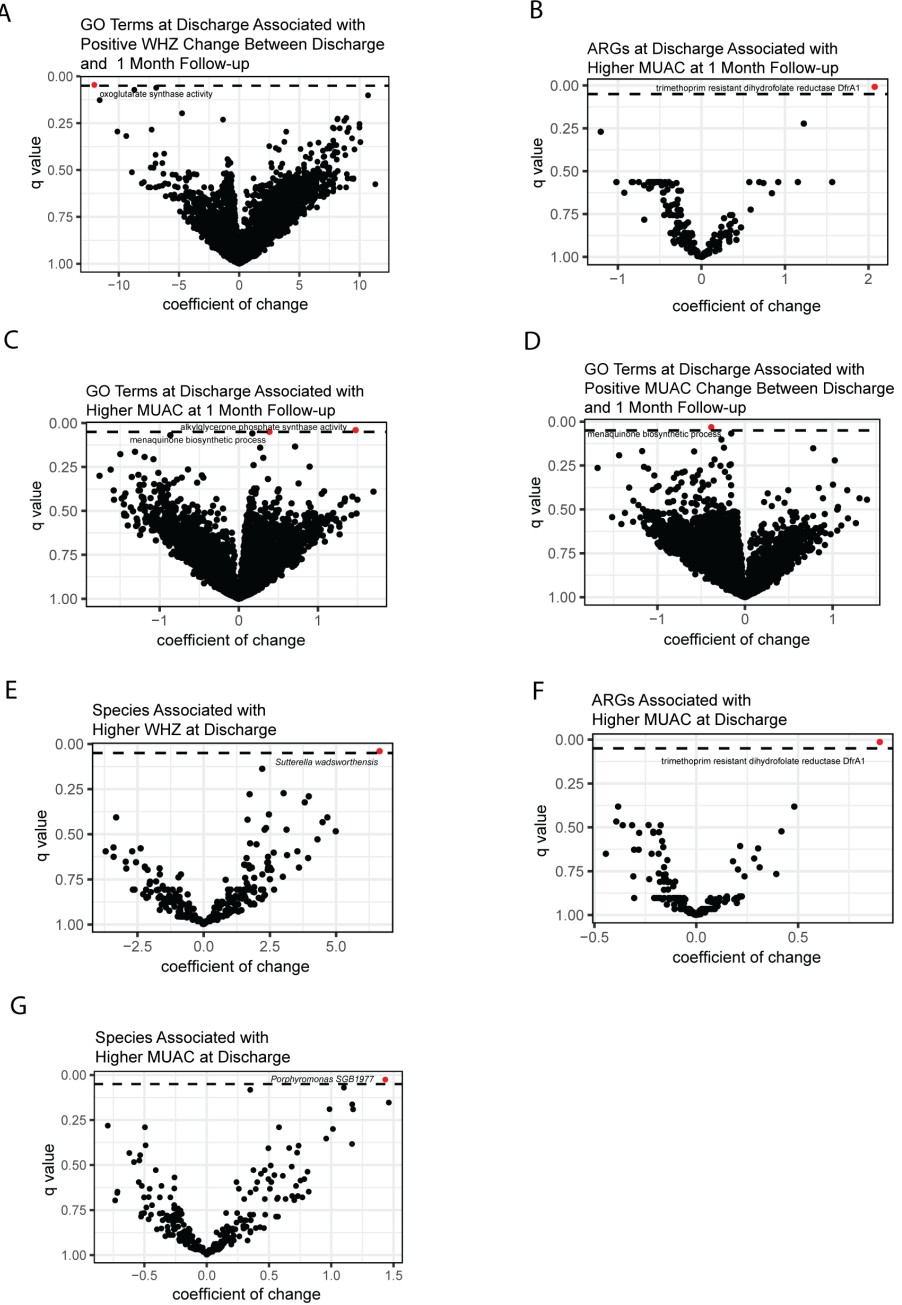

FIG 4 Specific microbiome features associated with differences in anthropometry. Coefficients of change of microbiome features correlated with the following anthropometric measurements, plotted against each feature's *q* value on an inverse y axis. Dotted line drawn at *q* < 0.05. Features ≤ 0.05 are highlighted in red. (A) Coefficients of change in microbiome functional potential at discharge defined by GO that are associated with positive change in WHZ between discharge and 1-month follow-up: oxoglutarate synthase activity (coeff: −12, SD: 3.3, *P* = 0.00043, *q* = 0.046). (B) Coefficients of change in microbiome ARGs at discharge that are associated with higher MUAC at 1-month follow-up: trimethoprim-resistant dihydrofolate reductase DfrA1 (coeff: 2.1, SD: 0.41, *P* = 3.5e-06, *q* = 0.0081). (C) Coefficients of change in microbiome functional potential at discharge defined by GO that are associated with higher MUAC at 1-month follow-up: menaquinone biosynthetic process (coeff: 0.39, SD 0.11, *P* = 0.00048, *q* = 0.050) and alkylglycerone phosphate synthase activity (coeff: 1.5, SD: 0.40, *P* = 0.00036, *q* = 0.040). (D) Coefficients of change in microbiome functional potential at discharge defined by GO that are associated with positive change in MUAC between discharge and 1-month follow-up: menaquinone biosynthetic process (coeff: −0.38, SD: 0.10, *P* = 0.00027, *q* = 0.032). (E) Coefficients of change in species relative abundance in the microbiome associated with higher WHZ at discharge: *Sutterella wadsworthensis* (coeff: 6.6, SD: 2.0, *P* = 0.0011, *q* = 0.040). (F) Coefficients of change in microbiome

Fig 4 (Continued)

ARGs associated with higher MUAC at discharge: trimethoprim-resistant dihydrofolate reductase DfrA1 (coeff: 0.90, SD: 0.20, *P* = 1.1e-05, *q* = 0.014). (G) Coefficients of change in species relative abundance in the microbiome associated with higher MUAC at discharge: *Porphyromonas SGB1977* (coeff: 1.43, SD: 0.40, *P* = 0.00057, *q* = 0.025).

were primarily identified for children of different sex, or of increasing age, both of which are normal to childhood microbiome development in both malnourished and non-malnourished children (16–19, 36, 37, 41). The duration of CMAM treatment did not have any effects on overall microbiome differences.

Our study has several important limitations. First, due to the limitations in the lack of additional sample collection over time, we were unable to determine if post-discharge relapse to AM could be related to the expansion or loss of specific bacterial species present in the microbiome at discharge. Similarly, we were unable to determine how much nutritional rehabilitation could have changed the microbiome of malnourished children between beginning CMAM, discharge, and post-discharge follow-up, as observed in other studies (23). Last, our study lacks a control group of children who have never experienced AM but have similar vulnerability to developing AM from the same community. As such, we were unable to determine how the microbiomes of children who relapse to AM may differ from non-malnourished children in the same geographic area. With these limitations in mind, these data suggest that treatment for uncomplicated SAM may transiently reduce the variance of the gut microbiome between treated children and that the assessment of the overall gut microbiome composition of children immediately following treatment may be insufficient when used alone as a predictive tool for AM relapse.

In addition to the immediate adverse consequences toward childhood growth and organ system development, AM also increases susceptibility to systemic and enteric infections, which is also being investigated in other children in our South Sudanese cohort (42). This vulnerability can be due to decreased immune responses from nutrient deficiencies, disruption of intestinal barriers, and/or the increased carriage of infectious pathogens within the gut (43). To address this, acutely malnourished children treated with CMAM guidelines are regularly prescribed amoxicillin (3). A study that investigated the effect of amoxicillin on ARG carriage in the microbiome of children with SAM found that although amoxicillin administration led to acutely elevated levels of ARGs in the gut, these effects were not sustained, and children treated with amoxicillin had improved gut microbiome maturation 2 years later compared to those who were treated with a placebo (17). This finding does not exclude the possibility that these transient increases in ARGs within the microbiome of children administered amoxicillin as a part of routine procedures may lead to acute infections from multi-drug-resistant organisms in the immediate post-treatment period, which may be driving relapses to AM. Relatedly, differences in ARG profiles may also predict the degree of sustained recovery post-CMAM treatment due to the incomplete modulation of the gut microbiome by antibiotics leading to increased susceptibility to AM relapse.

Our study found no relationship between ARG carriage at discharge and relapse to AM. Older age at discharge appears to affect the proportion of genetic content devoted toward beta-lactam ARGs, and children discharged with and without MAM have different distributions of ARGs. This suggests that there may be age-related exposures in South Sudan, such as increased cumulative exposure to antibiotics due to the high rate of AM relapse, that are leading to the retention of beta-lactam ARGs in the microbiome (44). The variations in ARG distribution between children discharged fully recovered or with MAM appear to be driven by ARG variations between groups that did not pass the feature prevalence filters in our linear mixed effect models. These variations could be driven either by modification of gut microbiome ARGs in response to antibiotics in children who fully recovered, or by the variations themselves, such as increased antibiotic resistance, which could be a driving factor for why children discharged with MAM did not achieve full WHO recovery with CMAM. Additional data points are required

to comment on the relationship between ARG distribution and full recovery. Overall, our results are limited by the lack of antibiotic exposure documentation. The records of the children whose exposure status was available were limited to whether the child was ever documented as being exposed to amoxicillin during their treatment for SAM, with missing information on the time of exposure relative to sample collection and the degree of antibiotic compliance. Due to these limitations, it is possible that we may have missed nuances in the changes in ARG quantity and profile that relate to time and length of exposure in our cohort. Furthermore, we cannot rule out whether children were ever exposed to amoxicillin outside of the clinical setting, including through environmental exposures such as proximity to livestock, given the high rates of cattle and animal ownership among families in this cohort (5, 45). We also cannot rule out whether children in our cohort were ever exposed to antibiotics other than amoxicillin in a relevant timeframe that may be affecting the distribution of ARGs. Despite these caveats, these limitations may reflect real conditions in LMICs, where antibiotic exposures are high, healthcare encounters are variable, and medication surveillance is low. Thus, despite these potential confounders, our data suggest that the quantity and distribution of ARGs likely hold low predictive value for predicting AM relapse in low-resource settings.

Our evaluation of microbiome functional potential also possesses several limitations. Estimates of functional potential through sequencing allow us to predict the potential metabolic pathways that bacteria may be able to activate in the gut, but do not indicate which pathways are active and affecting biology or AM. We did not observe any overall differences in microbiome potential between children who sustained recovery or those who relapsed to AM at 1-month follow-up. It is possible that treatment for SAM and the specific conditions experienced by children in our cohort may have reduced the variance of the metabolic potential of the bacteria between the microbiota of children at discharge, but these results need to be further investigated with metabolomics or other techniques (46, 47).

We found specific differences in the bacterial composition, ARGs, and functional potential of children at discharge that differentiate children with different anthropometric measurements and trajectories. These differences existed for MUAC and WHZ measurements, which are used to determine AM status, as well as for changes in WAZ and HAZ. Previous publications have indicated that microbiome dynamics can be correlated with growth trajectories and have moderate predictive value for linear growth (48, 49). Oxoglutarate synthase is an important regulator of bacterial metabolism that links carbon and nitrogen metabolic pathways (50). Decreases in its predicted potential activity in the microbiome at discharge were positively associated with higher WHZ at 1-month follow-up. Alkylglycerone phosphate synthase is important for the biosynthesis of membrane lipids; an increase in its predicted potential activity in the microbiome at discharge was positively associated with higher MUAC at 1-month follow-up (51). Finally, menaquinone biosynthesis is important in the bacterially mediated generation of vitamin K, and its predicted potential activity is both increased at discharge in children who have higher MUAC at 1-month follow-up, as well as decreased in children with greater positive changes in MUAC at 1-month follow-up (52). It is difficult to interpret the role of these associations in the context of AM due to the limitations in determining which organisms are driving these changes as well as complex bacterial metabolic interactions that cannot be resolved with our study methods. The observed contradictions in associations with predicted potential menaquinone biosynthetic pathway activity and anthropometric changes may be due to small errors in measurement.

Both higher MUAC at discharge and at 1-month follow-up were associated with increased trimethoprim-resistant dihydrofolate reductase activity in the gut microbiome at discharge. Trimethoprim is an antifolate antibacterial agent that is commonly combined with sulfamethoxazole under the name cotrimoxazole. Cotrimoxazole is listed as an essential medicine by South Sudan and is administered at every level of healthcare (53). The acquisition of trimethoprim-resistant dihydrofolate reductase activity in

the gut may be a direct result of increased cotrimoxazole exposure. Previous studies have examined the role of antibiotics on childhood growth in LMICs, with findings that indicated associations between antibiotic exposure and growth (54). Furthermore, a study investigating antibiotic stewardship in South Sudan revealed high levels of antibiotic misuse, including high levels of inappropriate prescription of cotrimoxazole (55). This suggests that children with increased healthcare encounters may be acquiring greater levels of trimethoprim-resistant dihydrofolate reductase and that the associations between this ARG and higher MUAC measurements may be directly from growth-promoting effects of antibiotic exposure or as a proxy for associations of increased healthcare access with higher anthropometry.

Increases in *Sutterella wadsworthensis* and *Porphyromonas SGB1977* in the microbiome of children at discharge were associated with higher WHZ or MUAC at discharge, respectively. *Sutterella wadsworthensis* has been identified as an important gut commensal in the control of dysbiosis by the human immune system, as well as the enhancement of host carbohydrate metabolism (56, 57). Furthermore, mouse models of nutrient restriction and enteric infection have also identified *S. wadsworthensis* as a protective bacterium in maintaining weight gain in subsequent nutrient restriction events (57). These characteristics all support the association of this bacterium with higher anthropometric measurements. The genus *Porphyromonas* has mainly been studied in the context of oral diseases and periodontitis. High levels of *Porphyromonas* in the gut have been associated with both healthy subjects and patients with gastric cancers. The association between *Porphyromonas SGB1977* and higher MUAC measurements could point to an unknown role of gut *Porphyromonas SGB1977* in childhood nutrition.

In summary, broad microbiome features of children treated for uncomplicated SAM at discharge are likely unable to predict relapse to AM in the short term. Specific microbiome changes may be associated with variations in anthropometry; however, additional investigations will be needed to validate the cause or effect of these observations. Among our study's findings, the majority of associations are driven by anthropometry measurements rather than categorical definitions of relapse. This is due to AM being defined by anthropometry cutoffs, which may result in a loss of nuance in nutritional trajectories reflected by changes in anthropometry below the AM threshold.

The goal of our study was to determine if microbiome signatures at discharge could predict relapse to AM, and it is important here to underscore the fact that a good diagnostic tool for AM should be reliable under a variety of conditions. Given the low-resource environments where AM is predominantly found, a tool that remains robust in these settings is necessary. Thus, although we acknowledge that our study possesses several limitations, we believe that there is not enough evidence that the microbiome alone can be used as a predictive tool for assessing risks of relapse to AM following treatment for uncomplicated SAM in low-resource settings. The joint interpretation of other predictive factors, such as access to water, sanitation, and hygiene, as well as other financial or household factors, may be needed to improve the predictive value of microbiome data to acute malnutrition relapse (13, 14, 58–60). Future studies employing additional timepoints of stool collection and rigorous documentation of clinical variables such as antibiotic administration will be necessary to determine the significance of the microbial composition, ARGs, and functional potential of the microbiome at discharge to the development of AM relapse.

## ACKNOWLEDGMENTS

We would like to thank all caregivers and children who generously provided their time, without whom this study would not have been possible. We acknowledge the significant contributions made by the Ministry of Health in South Sudan. Last, we give particular thanks to Ellyn Yakowenko, Bram Riems, Ronald Stokes-Walters, Dimple Save, David Gama Abugo, Jackson Lwate Hasan, and Lino Deng for their invaluable support of the study.

Conceptualization of this study was performed by K.Y., S.K., I.T., O.C., H.S., and D.J.S. Data curation was performed by K.Y., S.K., J.A., N.G.L., L.D.G., A.M., and H.S. Formal analysis, investigation, methodology, validation, and visualization were performed by K.Y. with supervision from D.J.S. Resources were contributed by K.Y., S.K., A.M., L.D.G., L.G., G.W., L.Z., N.G.L., J.A., D.S., M.G., K.A., A.M., I.T., O.C., H.S., and D.J.S. Technical support was provided by G.W. and L.Z. The original draft was written by K.Y. and S.K. All co-authors contributed to manuscript review and editing.

The U.S. Agency for International Development provided financial support for this article through the Bureau of Humanitarian Assistance. It was prepared under the terms of contract 720FDA19GR00278.The U.S. Agency for International Development provided financial support for this article through the Bureau of Humanitarian Assistance. It was prepared under the terms of contract 720FDA19GR00278.

## AUTHOR AFFILIATIONS

[1]Department of Pediatrics, Division of Infectious Diseases, Washington University School of Medicine in St. Louis, St. Louis, Missouri, USA

[2]Action Against Hunger, New York, New York, USA

[3]Feinstein International Center, Tufts University, Medford, Massachusetts, USA

[4]Environmental Health Group, Department of Disease Control, Faculty of Infectious and Tropical Diseases, London School of Hygiene and Tropical Medicine, London, United Kingdom

[5]Department of Infection Biology, Faculty of Infectious Diseases, London School of Hygiene and Tropical Medicine, London, United Kingdom

[6]Action Against Hunger, Juba, South Sudan

[7]Department of Pediatrics, University of Washington, Seattle, Washington, USA

[8]Department of Global Health, University of Washington, Seattle, Washington, USA

[9]Department of Epidemiology, University of Washington, Seattle, Washington, USA

[10]Department of Molecular Microbiology, Washington University School of Medicine in St. Louis, St. Louis, Missouri, USA

[11]Department of Obstetrics and Gynecology, Washington University School of Medicine in St. Louis, St. Louis, Missouri, USA

[12]Center for Women's Infectious Disease Research, Washington University School of Medicine in St. Louis, St. Louis, Missouri, USA

## AUTHOR ORCIDs

K. Yang http://orcid.org/0000-0002-5211-913X

L. Grignard http://orcid.org/0000-0001-8305-3981

J. Knee http://orcid.org/0000-0002-0834-8488

D. J. Schwartz http://orcid.org/0000-0003-1568-7733

## FUNDING

| Funder | Grant(s) | Author(s) |
| --- | --- | --- |
| National Institutes of Health | K08AI159384 | D. J. Schwartz |

## AUTHOR CONTRIBUTIONS

K. Yang, Conceptualization, Data curation, Formal analysis, Investigation, Methodology, Resources, Visualization, Writing – original draft | S. King, Conceptualization, Data curation, Funding acquisition, Investigation, Methodology, Resources, Writing – original draft, Writing – review and editing | A. Marshak, Data curation, Resources, Writing – review and editing | L. D'Mello-Guyett, Funding acquisition, Resources, Writing – review and editing | L. Grignard, Resources, Writing – review and editing | J. Knee, Resources, Writing – review and editing | G. Wong, Resources, Writing – review and editing | L. Zhao, Resources, Writing – review and editing | N. G. Lamaka, Resources, Writing – review and

editing | D. Save, Resources, Writing – review and editing | M. Gose, Resources, Writing – review and editing | A. Myers, Resources, Writing – review and editing | I. Trehan, Conceptualization, Resources, Writing – review and editing | O. Cumming, Conceptualization, Funding acquisition, Resources, Writing – review and editing | H. Stobaugh, Conceptualization, Data curation, Funding acquisition, Project administration, Resources, Supervision, Writing – review and editing | D. J. Schwartz, Conceptualization, Funding acquisition, Project administration, Resources, Supervision, Writing – review and editing

## DATA AVAILABILITY

All metagenomic sequencing is available at PRJNA1355224. Code used to generate figures and analyze statistical significance is included at https://github.com/DJSchwartzLab/AM-relapse-SSD.

## ETHICS APPROVAL

Ethical approval was obtained from the Solutions Institutional Review Board (#20200310), the London School of Hygiene and Tropical Medicine's (LSHTM) Research Ethics Committee (#18059), and the Ministry of Health of South Sudan (MOH/ERB6/2020).

## ADDITIONAL FILES

The following material is available online.

### Supplemental Material

**Data S1 (Spectrum03587-25-s0001.xlsx).** Metadata.
**Data S2 (Spectrum03587-25-s0002.xlsx).** Maaslin significant hits.
**Data S3 (Spectrum03587-25-s0003.xlsx).** Sequencing read counts.
**Supplemental figures (Spectrum03587-25-s0004.pdf).** Fig. S1–S17.

### Open Peer Review

**PEER REVIEW HISTORY (review-history.pdf).** An accounting of the reviewer comments and feedback.

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
