## [Reviewer comments · Microbiology Spectrum]

Microbiology Spectrum

Gut microbiome associations with acute malnutrition relapse in South Sudan

Kangning Yang, Sarah King, Anastasia Marshak, Lauren D'Mello-Guyett, Lynn Grignard, Jackie Knee, Galen Wong, Lingxia Zhao, Nancy Lamaka, Dimple Save, Mesfin Gose, Alesha Myers, Indi Trehan, Oliver Cumming, Heather Stobaugh, and Drew Schwartz

Corresponding Author(s): Drew Schwartz, Washington University in St Louis School of Medicine

Review Timeline:

Submission Date:	November 4, 2025
Editorial Decision:	January 17, 2026
Revision Received:	February 19, 2026
Accepted:	March 21, 2026

Editor: Steven Frese

Reviewer(s): The reviewers have opted to remain anonymous.

Transaction Report:

DOI: <https://doi.org/10.1128/spectrum.03587-25>

Re: Spectrum03587-25 (Gut microbiome associations with acute malnutrition relapse in South Sudan)

Dear Dr. Drew Joel Schwartz:

Thank you for the privilege of reviewing your work. Below you will find my comments, instructions from the Spectrum editorial office, and the reviewer comments. You should see Reviewer #2's comments in the attached.

Revision Guidelines

Sincerely,
Steven Frese
Editor
Microbiology Spectrum

Reviewer #1 (Comments for the Author):

Very difficult to follow the structure of this article - I'm used to seeing Introduction, Methods, Results, Discussion. The methods section begins on line 439 after results and discussion. The abstract also suffers from a lack of structure and does not identify the hypothesis of the study. As I understand it, children had rectal swabs taken at recovery from the index SAM episode, and then were placed into two categories based on anthropometry one month after sample collection: relapsed or not relapsed? Very difficult to follow the population used for this analysis
A flow chart to allow understanding of how the sample for this study was generated: Reference 5 cites 515 SAM children included in a 6 mo follow up period following recover from the index episode of SAM, of which 23 were lost to follow up and 4

died, leaving 488 children in the cohort.

However, this manuscript states that 490 children had a rectal swab collected - so a small discrepancy here that should be clarified

Of these 490 children with a rectal swab, a subset of 152 were subject to metagenomic analysis - with no explanation on selection criteria for these 152 children. How this subset of 152 was selected must be clearly explained.

Of the 152 with metagenomic analysis, 130 were discharged from SAM treatment meeting all criteria for recovery (WHZ > -2, MUAC > 125 mm and absence of oedema) while 22 were discharged while still classified as moderately acutely malnourished, presumably with a WHZ score between -3 and -2, but a MUAC > 125 mm as the South Sudan protocol provides for determining recovery based on the anthropometric criterion that qualified them for treatment (ie either MUAC or WHZ).

3. This manuscript refers the reader to reference 5 (King, S et al. Lancet Global Health Volume 13, Issue 1e98-e111 January 2025) to understand the cohort used for this analysis. On page 6 in the appendix of ref 5, the treatment protocol for South Sudan is given with SAM children being treated with RUTF; and SAM children being transferred to supplemental feeding programs for MAM once children improve from SAM to MAM. Children in supplemental feeding programs receive either fortified blended flour or lipid-based RUSF. The maximum duration of treatment for SAM children is listed as 6 months. It seems to me that duration of treatment with therapeutic or supplemental feeding could have an impact on modifications of the microbiome, but I do not see length of treatment addressed anywhere in this manuscript, nor determination of whether length of treatment (which apparently can extend to 6 months) is in anyway correlated with modifications of the microbiome.

4. Line 116 - implies between 61-75% of children were exposed to antibiotics. The definition of antibiotic exposure is not clear. Typically 100% of SAM children receive a 7 day course of Amoxicillin, so unclear why the number of children exposed to antibiotics is <100% unless there were problems with SAM treatment protocol adherence. Again length of treatment may intervene here: if children are in SAM/MAM treatment for up to 6 months, it is quite possible that they receive courses of antibiotics for intercurrent illness in addition to systematic amoxicillin over this fairly lengthy time period. Or maybe the authors are attempting to describe antibiotic treatment for SAM with concurrent infections (ie respiratory infection, diarrhea , urinary tract infection or skin infection?)

5. This manuscript requires major revision to clarify the focus of the analysis. Instead of getting bogged down in the complexities of different anthropometric definitions of recovery and relapse (WHZ vs MUAC), this attempt to describe differences in microbiome may benefit from a focus on those children who lost MUAC (Supplementary Figure 4) compared to those who did not one month post-discharge, as loss of MUAC may be more correlated with loss of muscle mass in this population of children than in decrease in WHZ.

Yang et al conducted an interesting and valuable study investigating children in South Sudan with malnutrition. This study investigates whether gut microbiome features at discharge can predict relapse to acute malnutrition (AM) among children recovering from severe acute malnutrition (SAM). Using metagenomic profiling of microbial composition, antimicrobial resistance genes, and predicted functional pathways, the authors assess associations between microbiome characteristics and relapse outcomes one-month post-recovery. It's clear that the authors have invested substantially in assembling and generating data from a challenging and clinically important cohort. Despite reporting largely negative results, this work contributes important information to the field and would become much more accessible and compelling to readers with some restructuring, clarification, and some additional pedagogical framing.

Below are comments aimed at improving clarity, consistency, and interpretation, as well as suggestions for strengthening the presentation of the results.

Line 38: The wording “discharged recovered” is unclear. Does this mean the children were discharged *because* they recovered? Consider rephrasing for clarity.

Line 45: *Sutterella* is misspelled here and throughout the manuscript. Additionally, the connection between antibiotic resistance genes and arm circumference is not immediately clear—some explanatory context would help.

Line 54: The phrase “presence of nutritional bilateral pitting oedema” could be made more accessible by adding a brief plain-language explanation.

Line 98: Why would ARGs be suspected in connection with relapse? In the above paragraphs, it mentions the prebiotics and antibiotics as part of treatment so maybe resistance results in less effective treatment? If so, it would be helpful to lay this out a bit more explicitly

Line 107: Please clarify the distinction between a rectal swab and a metagenomic sample in this study.

Line 109: Does “152 children sampled” refer to children sampled or children with samples that were successfully sequenced? Clarification is needed.

Line 113: Consider rephrasing to something like “deemed recovered at discharge based on X, Y, Z criteria,” and then parenthetically note “referred to as ‘discharged recovered’” to avoid awkward phrasing.

Line 118: The phrase “sustained relapsed” is unclear - do you mean “sustained recovery” vs “relapsed”?

Line 120: The contrast here is unclear - “as opposed to what?” MAM?

Lines 154-159: The paragraph describing the phylum-level distributions is difficult to follow as written. The heavy use of nested parentheses makes it hard for readers to track the groupings and presenting only two of the groups in Fig. 1D, while discussing all three, creates further misalignment. Restructure this section for readability. Also, Panel D in Figure 1 uses *MAM* as the comparison group, while the other panels use *AM*. It's unclear why different panels use different grouping variables. This becomes especially important because the three groups - sustained recovery, relapse, and "discharged from SAM but still having MAM" - do not appear to correspond to the same follow-up definitions. The third group seems not to have a 1-month follow-up classification, which is confusing.

Line 169: The summary at the end of this section may not be necessary and could be removed for better flow.

Line 178: The rationale for examining ARGs would benefit from further development here as well. Also, in this section, the comparisons shift to only "children discharged with or without AM," ignoring 1-month relapse status; although this is a valid comparison, the switch in grouping strategy between sections gives the manuscript an inconsistent flow. A clearer global explanation of the group definitions upfront - and why different analyses use different groupings - would be helpful.

Line 187: Why is amoxicillin singled out? A brief explanation would help contextualize its importance.

Line 197: The text is difficult to read due to many intercalated numerical values. Consider moving the numbers to the end of the sentence with a "respectively" statement.

Line 216: If interpreted correctly, could this also suggest that children discharged with AM may not have received sufficient antibiotics?

Line 223: Some of the numerical details could be moved into tables to avoid interrupting the narrative with multiple parenthetical statements.

Section starting at line 254: Would be helpful if the main significant findings shown in Fig. 4 were supplemented with a bit more information, including: **i)** Which species contributed to the significant pathways (oxoglutarate synthase, alkylglycerone phosphate synthase, menaquinone biosynthesis, etc.); **ii)** WHZ and MUAC appear repeatedly, yet the manuscript gives only minimal explanation of these measures. Including them in early figures or providing additional background would help orient readers; **iii)** If WHZ and MUAC are associated with pathways / ARGs while AM status is not, it would be valuable to discuss why anthropometric changes relate to microbiome features but clinical AM recovery does not; **iv)** Showing the underlying data (e.g., raw abundances or boxplots, can be a suppl. figure) behind the significant MaAsLin2 associations would make the findings more compelling than only presenting model coefficients; **v)** a figure summarizing the full significance distribution (e.g., volcano plots or ranked $-\log_{10} p$ -

values) would help readers see how the significant results stand out from the background.

Lines 276 and 285: Correct the spelling of *Sutterella*.

Line 314: The claim that treated children's microbiomes are "normalized" needs clarification. Normalized relative to what? There are no pre-treatment or healthy community controls presented.

Line 323: The fact that children are regularly treated with amoxicillin is important context and would be helpful to mention earlier, as it motivates the emphasis on amoxicillin-related ARGs.

Line 339: It remains unclear how antibiotic exposure influences both gut microbiome maturation and infection risk. If children with ongoing MAM have more ARGs, could resistance be contributing to prolonged MAM? This idea may warrant elaboration.

Line 342-343: The sentence "meeting the WHO standards for AM recovery ... may be changing the ARGs present in the microbiome" is not supported by the data, as only one microbiome timepoint was collected.

Sections on line 389 & 392: you write activity and do not include "potential". As is mentioned in the limitations section, "Estimates of functional potential through sequencing allows us to predict the potential metabolic pathways that bacteria may be able to activate in the gut, but do not indicate which pathways are active and affecting biology or AM."

Line 389: The phrase "accounted for with" reads awkwardly - consider rephrasing for clarity.

Line 403: This explanation is helpful. Bringing some of this context earlier would strengthen the rationale for focusing on ARGs throughout the paper.

General comments on figures: Some figure axes are not explained and there are an unnecessary number of digits in PCoA plot axes. There are a lot of box plots; consider adding the same color of the groups as in the PCoA; also, it would look cleaner to have the same font size on all the panels of a figure. In the boxplots there is a repetition in the title and y-axis which creates unnecessary text. In figure 4, make the species names italics, the categories in y axis are also a bit confusing. It's not clear if there are correlations or comparisons between groups, and, in that case, which group. Also, the addition of arrows below the plot would help the reader easily see that *Sutterella wadsworthensis* numbers are higher in group X or Y.

Dear reviewers,

We provide for your consideration a revision of our manuscript entitled “Gut microbiome associations with acute malnutrition relapse in South Sudan” which we are resubmitting to Microbiology Spectrum. We sincerely appreciate the opportunity to revise and resubmit our work. In our work to address reviewer critiques, we have made edits to clarify key points about our study cohort, elaborate on the motivation behind specific analyses, and streamline paragraphs and figures for consistency in nomenclature and readability. We are hopeful that these changes will address reviewer concerns and improve the clarity and value of our manuscript for readers of Microbiology Spectrum. These reviewer comments are included below along with our responses in bold.

Reviewer 1

Very difficult to follow the structure of this article - I'm used to seeing Introduction, Methods, Results, Discussion. The methods section begins on line 439 after results and discussion

We have moved the methods section so that it is now after the Introduction and before the Results.

The abstract also suffers from a lack of structure and does not identify the hypothesis of the study.

We have restructured the abstract to fit with Microbiology Spectrum’s abstract guidelines and clarified the hypothesis of the study.

As I understand it, children had rectal swabs taken at recovery from the index SAM episode, and then were placed into two categories based on anthropometry one month after sample collection: relapsed or not relapsed? Very difficult to follow the population used for this analysis.

The reviewer’s understanding of the study cohort is correct, and we appreciate the opportunity to clarify this important section for readers in our manuscript. To address this concern, we have added additional explanations of the study population in (Line 286 - 297) as well as in our methods (Line 129 - 143).

A flow chart to allow understanding of how the sample for this study was generated: Reference 5 cites 515 SAM children included in a 6 mo follow up period following recover from the index episode of SAM, of which 23 were lost to follow up and 4 died, leaving 488 children in the cohort.

However, this manuscript states that 490 children had a rectal swab collected - so a small discrepancy here that should be clarified.

We greatly appreciate the reviewer's request to expand upon the explanation of our cohort. The 490 children refer to a different group than the 488 children reported in the main results of King et al. Lancet Global Health (2025). Of the 515 children the reviewer is referencing from King et al. 2025, 490 had rectal swabs collected at the time of enrollment. In contrast, the 488 children cited in the main results represent those who completed the six-month follow-up period, excluding those who were lost to follow-up or died during the study. Notably, children were included in the main results of King et al. 2025 regardless of whether a rectal swab was taken at enrollment. Aside from taking acute malnutrition relapse data at the 1-month point from the King et al. 2025 paper, the population reported in our manuscript is standalone which is why we have not included a flow chart describing the larger cohort from the previous paper.

Of these 490 children with a rectal swab, a subset of 152 were subject to metagenomic analysis - with no explanation on selection criteria for these 152 children. How this subset of 152 was selected must be clearly explained.

We appreciate the reviewer giving us the opportunity to clarify this point. At the conception of this study, 152 children were selected for metagenomic analyses to create 1:1 matched pairs between relapsed and non-relapsed children at the 3-month follow-up time point. To capture a more relevant contribution of the gut microbiome to AM relapse, during the analysis phase the authors decided to use the 1-month follow-up time point instead. We have included explicit clarifications to this in our methods for clarity (Lines 132-137)

Of the 152 with metagenomic analysis, 130 were discharged from SAM treatment meeting all criteria for recovery (WHZ >-2, MUAC > 125 mm and absence of oedema) while 22 were discharged while still classified as moderately acutely malnourished, presumably with a WHZ score between -3 and -2, but a MUAC > 125 mm as the South Sudan protocol provides for determining recovery based on the anthropometric criterion that qualified them for treatment (ie either MUAC or WHZ).

The reviewer's understanding of this population is correct. A subset of children discharged from CMAM remained in a state of MAM by the WHO AM definition due to being < -2 WHZ despite having a MUAC at >125. We used this subset of children to investigate gut microbiome differences that may exist between children with MAM vs those who recover fully per WHO guidelines following CMAM treatment. We have clarified this in our text and methods (Lines 137 - 143, 285 - 297)

3. This manuscript refers the reader to reference 5 (King, S et al. Lancet Global Health Volume 13, Issue 1e98-e111 January 2025) to understand the cohort used for this analysis. On page 6 in the appendix of ref 5, the treatment protocol for South Sudan is given with SAM children being treated with RUTF; and SAM children being transferred to supplemental feeding programs for MAM once children improve from SAM to MAM. Children in supplemental feeding programs receive either fortified blended flour or lipid-based RUSF. The maximum duration of treatment for SAM children is listed as 6 months. It seems to me that duration of treatment with therapeutic or supplemental feeding could have an impact on modifications of the microbiome, but I do not see length of treatment addressed anywhere in this manuscript, nor determination of whether length of treatment (which apparently can extend to 6 months) is in anyway correlated with modifications of the microbiome.

We appreciate the reviewer's insight into the contribution of the length of treatment as a variable that may be affecting changes in gut microbiome features of children treated for SAM. The length of treatment is included in our metadata as "in_tot_los" (supplementary file 1). Length of stay did not impact the species richness, Shannon diversity, number of ARGs, nor the number of potential functional pathways at discharge in our cohort (supplementary figures 1C-D, 6C-D, 8E-F, 9C-D, 14C-D). We have included these analyses in our text (Lines 323-325, 384-385, 400-401, 411-412, 458-460).

4. Line 116 - implies between 61-75% of children were exposed to antibiotics. The definition of antibiotic exposure is not clear. Typically 100% of SAM children receive a 7 day course of Amoxicillin, so unclear why the number of children exposed to antibiotics is <100% unless there were problems with SAM treatment protocol adherence. Again length of treatment may intervene here: if children are in SAM/MAM treatment for up to 6 months, it is quite possible that they receive courses of antibiotics for intercurrent illness in addition to systematic amoxicillin over this fairly lengthy time period. Or maybe the authors are attempting to describe antibiotic treatment for SAM with concurrent infections (ie respiratory infection, diarrhea , urinary tract infection or skin infection?).

We appreciate the reviewer's insight and knowledge of this context. The reviewer is correct, under normal CMAM protocol, 100% of SAM children would have been expected to receive amoxicillin. The "amoxicillin exposure" data in our study is based on clinical records, mainly, whether or not a nurse specifically noted the exposure in a child's chart. Therefore, there are several scenarios that could apply to these children that were originally referred to as "not exposed" in our cohort. 1) These children received amoxicillin, but the nurse did not check the box in their records resulting in a documentation oversight 2) supply chain breaks in South Sudan created actual periods where children were NOT exposed to antibiotics. Unfortunately, we are unable to confirm which is the case, nor do we have insight into

antibiotic adherence in this population, however we agree with the reviewer that it is incorrect to label these children as having no amoxicillin exposure. We have made changes to Table 1 and (Lines 302-309) to redefine these children as having documented or no documentation of antibiotic exposure instead of exposed vs not exposed.

5. This manuscript requires major revision to clarify the focus of the analysis. Instead of getting bogged down in the complexities of different anthropometric definitions of recovery and relapse (WHZ vs MUAC), this attempt to describe differences in microbiome may benefit from a focus on those children who lost MUAC (Supplementary Figure 4) compared to those who did not one month post-discharge, as loss of MUAC may be more correlated with loss of muscle mass in this population of children than in decrease in WHZ.

We thank reviewer 1 for their comments and constructive criticism. We have made several edits to the manuscript to streamline our analyses for clarity. Reviewer 1's comments about the importance of MUAC loss is well taken. Change in MUAC is one of the several indices that we looked at throughout the paper, and we have edited both the figure legends and the text to make it easier for readers to evaluate this variable.

Our decision to provide a general view of how different anthropometric indices may be correlated with changes in microbiome features is based on the fact that that AM recovery in CMAM treatment is definitional rather than biological and to evaluate anthropometric trajectories as a proxy for nutritional status or risk factor for AM relapse outside of these definitional categories or below the definitional cutoffs. We are unable to pinpoint exactly which anthropometric measures would be most relevant for our study and have included several anthropometric measurements to provide a more comprehensive understanding of how microbiome differences may have impacted our study population.

Reviewer 2

Yang et al conducted an interesting and valuable study investigating children in South Sudan with malnutrition. This study investigates whether gut microbiome features at discharge can predict relapse to acute malnutrition (AM) among children recovering from severe acute malnutrition (SAM). Using metagenomic profiling of microbial composition, antimicrobial resistance genes, and predicted functional pathways, the authors assess associations between microbiome characteristics and relapse outcomes one-month post-recovery. It's clear that the authors have invested substantially in assembling and generating data from a challenging and clinically important cohort. Despite reporting largely negative results, this work contributes important information to the field and would become much more accessible and compelling to readers with some restructuring, clarification, and some additional pedagogical framing.

Below are comments aimed at improving clarity, consistency, and interpretation, as well as

suggestions for strengthening the presentation of the results.

We would like to thank the reviewer for their comprehensive evaluation of our manuscript and their kind comments and thoughtful critiques. We believe that implementing the reviewer's suggestions have made a positive impact on the readability and clarity of our study and we appreciated this opportunity to streamline our manuscript. Please see our changes implemented below.

Line 38: The wording “discharged recovered” is unclear. Does this mean the children were discharged *because* they recovered? Consider rephrasing for clarity.

We agree with the reviewer that “discharged recovered” is ambiguous. Our intention here was to point to the 130 children that were discharged as “fully recovered” (defined later in the text) out of the 152 children in our cohort that we use to look at 1 month AM relapse. We have restructured our abstract to reflect Microbiology Spectrum's format for abstracts and removed this confusing sentence.

Line 45: *Sutterella* is misspelled here and throughout the manuscript. Additionally, the connection between antibiotic resistance genes and arm circumference is not immediately clear-some explanatory context would help

We would like to thank the reviewer for catching this mistake. We have fixed all instances of this typo in our manuscript. We have restructured our abstract to reflect Microbiology Spectrum's format for abstracts and added additional text to frame MUAC as a proxy for health in this context (Lines 39-40)

Line 54: The phrase “presence of nutritional bilateral pitting oedema” could be made more accessible by adding a brief plain-language explanation.

We have added a brief plain-language explanation (Lines 67-69)

Line 98: Why would ARGs be suspected in connection with relapse? In the above paragraphs, it mentions the prebiotics and antibiotics as part of treatment so maybe resistance results in less effective treatment? If so, it would be helpful to lay this out a bit more explicitly.

We thank the reviewer for this suggestion. Antibiotics (specifically amoxicillin) are a routine part of CMAM treatment in South Sudan, and we have added additional background in the Introduction (Lines 75, 92-94), additional framing in the Results (Lines 371-378 ,426-430), and additional commentary in the Discussion (Lines 551-554, 563-566), to allow readers to better understand our rationale in analyzing ARGs in the microbiome.

Line 107: Please clarify the distinction between a rectal swab and a metagenomic sample in this study.

All metagenomic samples are rectal swabs, but not all rectal swabs collected were metagenomically sequenced. We have edited our text to lay this out more clearly in the Methods (Lines 131-132, 200-201) and in the Results (Lines 282-285)

Line 109: Does “152 children sampled” refer to children sampled or children with samples that were successfully sequenced? Clarification is needed.

152 refers to both the intended number of children in our study cohort as well as the number of successfully sequenced samples, as every rectal swab selected for this study was successfully sequenced, we have included this line in the methods (Lines 131-132, 200-201). As with the above comment, we have edited our text to provide the readers with a better understanding of our rationale and procedures in sampling and cohort construction. (Lines 282-285)

Line 113: Consider rephrasing to something like “deemed recovered at discharge based on X, Y, Z criteria,” and then parenthetically note “referred to as ‘discharged recovered’” to avoid awkward phrasing.

We have changed our text to this phrasing (Line 287)

Line 118: The phrase “sustained relapsed” is unclear - do you mean “sustained recovery” vs “relapsed”?

Yes, we intended the sentence to compare those who sustained recovery compared to those who relapsed at 1 month follow-up. This has been corrected (Line 304)

Line 120: The contrast here is unclear - “as opposed to what?” MAM?

We intended the sentence to mean that children who had sustained recovery at 1 month follow-up had higher anthropometric indices at discharge than those who relapsed to AM at 1 month follow-up. We have amended the text to explicitly call out this comparison (Lines 312-313).

Lines 154-159: The paragraph describing the phylum-level distributions is difficult to follow as written. The heavy use of nested parentheses makes it hard for readers to track the groupings and presenting only two of the groups in Fig. 1D, while discussing all three, creates further misalignment. Restructure this section for readability. Also, Panel D in Figure 1 uses *MAM* as the comparison group, while the other panels use *AM*. It’s unclear why different panels use different grouping variables. This becomes especially important because the three groups - sustained recovery, relapse, and “discharged from SAM but still having MAM” - do not appear to

correspond to the same follow-up definitions. The third group seems not to have a 1-month follow-up classification, which is confusing.

We appreciate the reviewer's suggestions about readability and consistency in this section and others. We have amended the paragraph describing phylum-level distributions into smaller paragraphs for readability and removed the use of nested parentheses throughout the entire document (Lines 348-351, 354-355). We have also edited all figure legends for consistency in naming.

The intention of this study was to look at features of the microbiome at discharge that may predict relapse to acute malnutrition in the post treatment window, a portion of children were discharged with MAM due to having a MUAC > 125mm but a WHZ < -2. Since this group is already starting off as acutely malnourished, we did not use this group of children for any analyses involving 1 month follow-up relapse data. We only used this "discharged with MAM" group to look at differences that may exist between children who were discharged "fully recovered" according to the WHO standards of AM recovery and children who were discharged with MAM. We have further clarified the separation and usage of these 3 groups in the main text (Lines 286-297) and in the Methods (Lines 137-143).

Line 169: The summary at the end of this section may not be necessary and could be removed for better flow.

We thank the reviewer for this suggestion and we have condensed this paragraph into a higher-level summary of the section (Lines 365-368).

Line 178: The rationale for examining ARGs would benefit from further development here as well. Also, in this section, the comparisons shift to only "children discharged with or without AM," ignoring 1-month relapse status; although this is a valid comparison, the switch in grouping strategy between sections gives the manuscript an inconsistent flow. A clearer global explanation of the group definitions upfront - and why different analyses use different groupings - would be helpful

Similar to the comment on Line 154-159, we have added additional explanations of our study groups (Lines 137-143, 282-285). We have also added specific context in this section for the rationale of examining ARGs with our different groups using the reviewer's suggestion that ARGs may be driving the completeness of recovery post CMAM (full recovery vs discharged with MAM) and/or if the recovery is sustained in the post treatment period (Lines 371-378). We have also expanded on this hypothesis in our interpretation of the results in the Discussion (Lines 547-554, Lines 563-566)

Line 187: Why is amoxicillin singled out? A brief explanation would help contextualize its importance.

Amoxicillin is singled out as it is antibiotic of choice given in South Sudan with CMAM treatment. All documented antibiotic administration mentioned in our study with our cohort refer exclusively to amoxicillin. We agree with the reviewer that this point was not made apparent in our original text, and we have amended our document and table 1 to reference amoxicillin specifically when antibiotic administration is mentioned.

Line 197: The text is difficult to read due to many intercalated numerical values. Consider moving the numbers to the end of the sentence with a “respectively” statement.

We have amended all sentences with intercalated numerical values into smaller sentences with numbers in parentheses at the end.

Line 216: If interpreted correctly, could this also suggest that children discharged with AM may not have received sufficient antibiotics?

We thank the reviewer for providing this interesting perspective and we have added this suggestion as an additional interpretation of results in the text (Lines 427-430, 563-566)

Line 223: Some of the numerical details could be moved into tables to avoid interrupting the narrative with multiple parenthetical statements.

Similar to the above comments, we have amended all sentences with intercalated numerical values into smaller sentences with numbers in parentheses at the end.

Section starting at line 254: Would be helpful if the main significant findings shown in Fig. 4 were supplemented with a bit more information, including: i) Which species contributed to the significant pathways (oxoglutarate synthase, alkylglycerone phosphate synthase, menaquinone biosynthesis, etc. ; ii) WHZ and MUAC appear repeatedly, yet the manuscript gives only minimal explanation of these measures. Including them in early figures or providing additional background would help orient readers; iii) If WHZ and MUAC are associated with pathways / ARGs while AM status is not, it would be valuable to discuss why anthropometric changes relate to microbiome features but clinical AM recovery does not; iv) Showing the underlying data (e.g., raw abundances or boxplots, can be a suppl. figure) behind the significant MaAsLin2 associations would make the findings more compelling than only presenting model coefficients; (show individual figures)

v) a figure summarizing the full significance distribution (e.g., volcano plots or ranked $-\log_{10} p$ values) would help readers see how the significant results stand out from the background.

We greatly appreciate all of the suggestions from the reviewer in this section. We were unable to attribute significant pathways to any specific species with confidence, and we amended the text to more clearly include this limitation in this section (Lines 485 - 486).

We agree with the reviewer that it is important to better contextualize the usage of WHZ and MUAC since it makes up a large portion of our analysis. We have also included a more thorough description of the metrics in the Introduction (Lines 61-65), as well as added an explanation for looking at associations between microbiome features and anthropometry in the main text (Lines 287-301). We have also added additional interpretation of our analysis in the Discussion for why anthropometry was associated with specific microbiome features while AM relapse at 1 month follow-up was not (Lines 506-508, 645-648).

We appreciate the reviewer's suggestions regarding data visualization in this section and have taken this opportunity to revise our figure 4 into separate volcano plots for each feature type and anthropometric measurement where significant values were present. We have also added a supplementary figure with the MaAsLin2 output graphs for each significant feature showing the raw associations of each feature to each anthropometric value.

Lines 276 and 285: Correct the spelling of *Sutterella*.

We have corrected all misspellings of *Sutterella* in our text.

Line 314: The claim that treated children's microbiomes are "normalized" needs clarification. Normalized relative to what? There are no pre-treatment or healthy community controls presented.

We agree with the reviewer that the term normalized is misused here. We have amended the text from normalized to "reduce the variance of the gut microbiome between treated children" as this was the original intent of this statement (Line 533)

Line 323: The fact that children are regularly treated with amoxicillin is important context and would be helpful to mention earlier, as it motivates the emphasis on amoxicillin-related ARGs.

We have amended our text to introduce amoxicillin administration earlier on in the Introduction (Lines 302, 371).

Line 339: It remains unclear how antibiotic exposure influences both gut microbiome maturation and infection risk. If children with ongoing MAM have more ARGs, could resistance be contributing to prolonged MAM? This idea may warrant elaboration.

We appreciate the reviewer's insight and incorporated this interpretation in several places in our Results and Discussion (Lines 371-378, 427-430, 547-554, 563-566).

Line 342-343: The sentence "meeting the WHO standards for AM recovery ... may be changing the ARGs present in the microbiome" is not supported by the data, as only one microbiome timepoint was collected.

We have deleted this sentence

Sections on line 389 & 392: you write activity and do not include “potential”. As is mentioned in the limitations section, “Estimates of functional potential through sequencing allows us to predict the potential metabolic pathways that bacteria may be able to activate in the gut, but do not indicate which pathways are active and affecting biology or AM.”

We have amended all mentions of “activity” to include “potential”

Line 389: The phrase “accounted for with” reads awkwardly - consider rephrasing for clarity.

We have replaced the phrase “accounted for with” with “resolved” for clarity

Line 403: This explanation is helpful. Bringing some of this context earlier would strengthen the rationale for focusing on ARGs throughout the paper.

Similar to the above comments, we have amended our document to include explicit mentions of amoxicillin administration further up in the text (Lines 302, 371)

General comments on figures: Some figure axes are not explained and there are an unnecessary number of digits in PCoA plot axes. There are a lot of box plots; consider adding the same color of the groups as in the PCoA; also, it would look cleaner to have the same font size on all the panels of a figure normalize font size. In the boxplots there is a repetition in the title and y-axis which creates unnecessary text. In figure 4, make the species names italics, the categories in y axis are also a bit confusing. It's not clear if there are correlations or comparisons between groups, and, in that case, which group. Also, the addition of arrows below the plot would help the reader easily see that *Sutterella wadsworthensis* numbers are higher in group X or Y.

We thank the reviewer for their suggestions and appreciate the opportunity to streamline our figures for better visual interpretation. We have decreased the number of sig figs across our text and our figures as well as given each group a colour for the box plots that stay consistent throughout the entire paper. We have also standardized the font and font sizes across all figure labels, italicized all instances of bacterial species, and decreased unnecessary and redundant text in the figure titles and axes. We have completely remade figure 4 to allow readers to more easily see the direction and significance of the significant associations in relation to the anthropometric measurements mentioned.

Re: Spectrum03587-25R1 (**Gut microbiome associations with acute malnutrition relapse in South Sudan**)

Dear Dr. Drew Joel Schwartz:

Thank you for your patience!

Your manuscript has been accepted, and I am forwarding it to the ASM production staff for publication. Your paper will first be checked to make sure all elements meet the technical requirements. ASM staff will contact you if anything needs to be revised before copyediting and production can begin. Otherwise, you will be notified when your proofs are ready to be viewed.

Sincerely,
Steven Frese
Editor
Microbiology Spectrum

Reviewer #2 (Comments for the Author):

The authors have responded constructively and have improved the clarity and framing of the manuscript. The revisions clarifying cohort definitions, sampling strategy, and the rationale for examining ARGs are helpful, and the revised figures (particularly Fig. 4) improve the presentation of the association analyses. The authors have also moderated several interpretations and clarified the distinction between predicted functional potential and microbial activity. One additional point worth noting, which was not raised in the initial review, is that the analyses presented primarily evaluate associations rather than predictive models; accordingly, the conclusions regarding predictive value are best interpreted in that context.